# SynthWorlds: Controlled Parallel Worlds for Disentangling Reasoning and Knowledge in Language Models

**Ken Gu**[1][*], **Advait Bhat**[1], **Mike A. Merrill**[2], **Robert West**[3]
**Xin Liu**[4], **Daniel McDuff**[4], **Tim Althoff**[1]
[1]University of Washington, [2]Stanford University, [3]EPFL, [4]Google Research
⯃ https://github.com/behavioral-data/synthworlds
🤗 https://huggingface.co/datasets/kenqgu/synthworlds

## ABSTRACT

Evaluating the reasoning ability of language models (LMs) is complicated by their extensive parametric world knowledge, where benchmark performance often reflects factual recall rather than genuine reasoning. Existing datasets and approaches (e.g., temporal filtering, paraphrasing, adversarial substitution) cannot cleanly separate the two. We present SYNTHWORLDS, a framework that disentangles task reasoning complexity from factual knowledge. In SYNTHWORLDS, we construct parallel corpora representing two worlds with identical interconnected structure: a real-mapped world, where models may exploit parametric knowledge, and a synthetic-mapped world, where such knowledge is meaningless. On top of these corpora, we design two mirrored tasks as case studies: multi-hop question answering and page navigation, which maintain equal reasoning difficulty across worlds. Experiments in parametric-only (e.g., closed-book QA) and knowledge-augmented (e.g., retrieval-augmented) LM settings reveal a persistent *knowledge advantage gap*, defined as the performance boost models gain from memorized parametric world knowledge. Knowledge acquisition and integration mechanisms reduce but do not eliminate this gap, highlighting opportunities for system improvements. Fully automatic and scalable, SYNTHWORLDS provides a controlled environment for evaluating LMs in ways that were previously challenging, enabling precise and testable comparisons of reasoning and memorization.

## 1 INTRODUCTION

Language model (LM) agents are increasingly expected to autonomously complete complex tasks that require retrieve new information, reason over it, and synthesize novel insights. These capabilities underpin emerging applications such as web navigation, where agents need to traverse linked information to locate relevant content (Ning et al., 2025); personal health insights, where they must connect medical data with external resources to inform advice (Heydari et al., 2025); and scientific discovery, where it is necessary to integrate findings scattered across research articles to form new hypotheses (Yamada et al., 2025). Success in these settings requires operating over richly structured knowledge environments, navigating interlinked documents, resolving indirect references, and integrating evidence spread across multiple sources.

Yet, as LMs continue to be trained on massive web corpora (often with undisclosed training data), it remains unclear to what extent their performance reflects genuine reasoning versus the reciting of memorized knowledge (Carlini et al., 2023; Wu et al., 2024). Many benchmark tasks depend on *factual world knowledge* models likely encountered during training (Sainz et al., 2023; Xu et al., 2024b; Zhou et al., 2023). This undermines two goals: scientifically, it prevents isolating reasoning ability (i.e., functional linguistic competence) from memorization (i.e., formal linguistic competence) (Mahowald et al., 2024; Lu et al., 2024); practically, it limits confidence in deploying systems to novel environments (i.e., scientific discovery).

---

[*]Correspondence to kenqgu@cs.washington.edu

To distinguish reasoning from reciting, researchers have explored several strategies. One approach is manual curation of "*clean*" evaluation sets, which provides novelty but is costly, difficult to scale, and requires continual updates. For example, ToolQA (Zhuang et al., 2023), a benchmark released in 2023 to distinguish between questions answerable from an LM's internal knowledge and those requiring external information, included GSM8K questions derived from "*error cases made by ChatGPT*" at the time. However, subsequent work has shown that newer LMs may already memorize many of these answers (Zhang et al., 2024; Mirzadeh et al., 2025). Another approach, synthetic dataset generation (Huang et al., 2025; Hsieh et al., 2024), promises scalability, but often involves using existing content directly (e.g., novels) and thereby results in parametric knowledge leakage or relies on overly simplistic templates (e.g., "*The job of David is a farmer. The hobby of David is birdwatching.*"), limiting their ability to probe reasoning in realistic, richly interconnected settings.

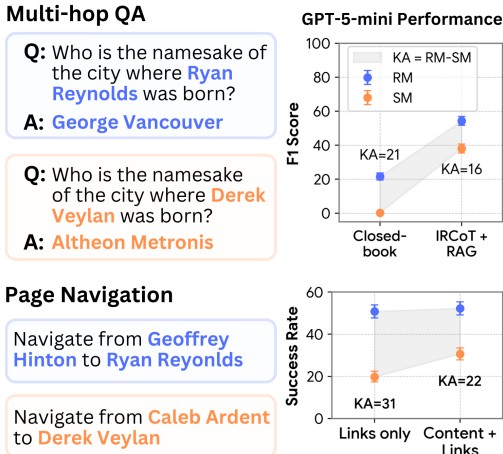

**Multi-hop QA**

**Q:** Who is the namesake of the city where **Ryan Reynolds** was born?

**A:** George Vancouver

**Q:** Who is the namesake of the city where **Derek Veylan** was born?

**A:** Altheon Metronis

**Page Navigation**

Navigate from **Geoffrey Hinton** to **Ryan Reyonlds**

Navigate from **Caleb Ardent** to **Derek Veylan**

Figure 1: **Controlled experiments from SYNTHWORLDS corpora.** We measure the *knowledge advantage gap* (KA) as the performance difference between parallel tasks *mapped* to **real-world** (RM) and **synthetic** (SM) entities. Retrieval and page content boosts performance but the gap persists.

Crucially, evaluations based *only* on synthetic unseen tasks still leave open questions about performance. Success demonstrates reasoning in isolation, but it does not reveal how much models typically rely on prior knowledge as a scaffold. Failure, on the other hand, is ambiguous: the reasoning chain underlying the task may be too difficult for models to succeed, or the model may simply lack the background knowledge it usually exploits. Without controlling both task difficulty and requirements for parametric knowledge, such evaluations leave the contributions of reasoning and memorization entangled.

To address this, we introduce SYNTHWORLDS, a framework for disentangling reasoning from factual knowledge. Parallel synthetic corpora are constructed to represent different *worlds* that replicate the structure and complexity of real-world information ecosystems. One corpus is mapped to **real-world** entities (e.g. *Geoffrey Hinton*), while the other is mapped to **synthetic** entities (e.g. *Caleb Ardent*), thereby obscuring the usefulness of parametric knowledge. This design allows us to quantify the *knowledge advantage gap* (i.e., the performance difference between real-mapped **[RM]** and synthetic-mapped **[SM]** settings) and to evaluate how knowledge acquisition methods (e.g., providing page content, retrieval-augmented generation) and integration strategies (e.g., chain-of-thought prompting, agentic reasoning) impact this gap (Fig. 1). The gap clarifies to what extent models rely on reasoning versus recall, and whether augmentation substitutes for or amplifies prior knowledge.

To support comparisons at scale, SYNTHWORLDS automatically generates parallel corpora from triplet facts in a knowledge graph (§3). To obscure factual knowledge, entities are renamed with surface-form–consistent transformations that preserve both type and name-derivation consistency before rendering facts into documents (Agarwal et al., 2021; Josifoski et al., 2023). Specifically, people receive person names (Geoffrey Hinton → Caleb Ardent), cities receive city names (Toronto → Metrovale), and derived names maintain consistency (University of Toronto → University of Metrovale, not University of Grandvale). This process yields corpora with identical reasoning structures while removing familiarity with entity-specific facts, resulting in coherent worlds where tasks require reasoning over complex documents under controlled relevance of parametric knowledge.

To demonstrate the utility of our SYNTHWORLDS framework, we generate two parallel corpora derived from Wikidata: SYNTHWORLD-RM and SYNTHWORLD-SM (§4). On top of each corpus, we construct two reasoning-intensive tasks as case studies: multi-hop question answering (QA) (Trivedi et al., 2022; Ho et al., 2020) and page navigation (West & Leskovec, 2012) with fine-grained control over difficulty.

In our experiments, we evaluate LMs on these tasks to quantify the knowledge advantage gap, first in settings where models rely only on parametric knowledge (closed-book QA for multi-hop reason-

ing and page names only for navigation), and then under conditions where knowledge augmentation (retrieval for QA, access to page contents for navigation) and integration strategies (e.g., chain-of-thought prompting) are provided (§5). Across both tasks, we find clear performance gaps between real-mapped and synthetic-mapped settings. While knowledge integration improves performance in both cases (and in some instances narrows the gap), the gap persists. This persistence highlights opportunities for future work to design more effective knowledge integration schemes and to systematically study system behavior when models encounter novel environments (§6). We contribute:

1. **A scalable framework for generating rich, interconnected corpora and tasks** that disentangle and task reasoning difficulty from parametric knowledge.

2. **Two parallel corpora with corresponding task datasets.** We instantiate the SYNTH-WORLDS framework with SYNTHWORLD-RM and SYNTHWORLD-SM, paired at the document, fact, and task levels to enable controlled evaluation. Each corpus contains 6,920 documents covering 161K facts, along with 1.2K multi-hop QA and 1K page navigation instances. To support future research, we release these resources publicly.

3. **An empirical analysis of LMs across parametric-only and knowledge-augmented settings** using our parallel datasets to quantify the knowledge advantage gap, which prior setups do not fully isolate. Our analysis reveals persistent shortcomings even with knowledge augmentation.

## 2 RELATED WORK

**Human Curated Data for Reasoning Evaluation.** As LM capabilities continue to improve and become widely deployed, researchers have relied on manually curated benchmarks to evaluate reasoning in settings not already covered by training data (Kazemi et al., 2025; Wei et al., 2025; Hendrycks et al., 2021; Cobbe et al., 2021; Bean et al., 2024; Srivastava et al., 2023; Tang & Yang, 2024; SU et al., 2025). These benchmarks are effective when first released but grow less informative over time as time passes. For example, MuSiQue (Trivedi et al., 2022), released in 2021 as a multi-hop QA benchmark, was originally designed to contain questions that models could not answer without the reference text. Despite this intent, it is still used across many evaluations today (Li et al., 2024; Zhang et al., 2025; Gutiérrez et al., 2025), even though current LMs (e.g., Llama-3.3-70B) achieve over 26% F1 score on these questions without any documents (Gutiérrez et al., 2025). This makes it difficult to assess whether improved performance reflects genuine advances in reasoning and retrieval capabilities that would be informative of systems deployed in unseen environments. As a result, researchers must continually spend effort to construct new datasets and tasks (Gu et al., 2024; Tang & Yang, 2024; Monteiro et al., 2024; Bai et al., 2025). These efforts require substantial expertise, grow increasingly complex as models advance, and are slow and costly to scale. In contrast, SYNTHWORLDS introduces a scalable framework to construct complex text data and associated reasoning tasks, reducing the manual curation burden while maintaining evaluation quality.

**Synthetic/Perturbed Data for Reasoning Generalization.** Given the resources needed to build high-quality human-generated data, researchers have developed methods to compose synthetic data or introduce perturbations to evaluate the reasoning generalization of LMs (Huang et al., 2025; Wu et al., 2024; Levy et al., 2024; Hsieh et al., 2024; Gu et al., 2025). These approaches reveal important weaknesses when LMs are tested outside familiar conditions or over long contexts, but they do not disentangle reasoning ability from reliance on parametric factual knowledge. Other efforts address this separation more directly, for example by focusing on real-time factual updates (Kasai et al., 2023; Vu et al., 2024) or by generating synthetic text (Gong et al., 2025; Allen-Zhu & Li, 2024; Monea et al., 2024). However, such work typically targets narrow aspects of knowledge or simplifies away the complexity and interconnectedness of real-world corpora, making it difficult to generalize findings to realistic scenarios (e.g., web navigation). Our work complements these lines by isolating the independent impacts of LM reasoning and parametric factual knowledge on task performance. Through controllable parallel dataset construction, we enable precise measurement of the *knowledge advantage gap* across common LM settings (e.g., in-context learning, RAG, agentic workflows), analysis on how different forms of knowledge augmentation influence this gap, and direct comparisons of model behavior in *novel* versus *familiar* settings.

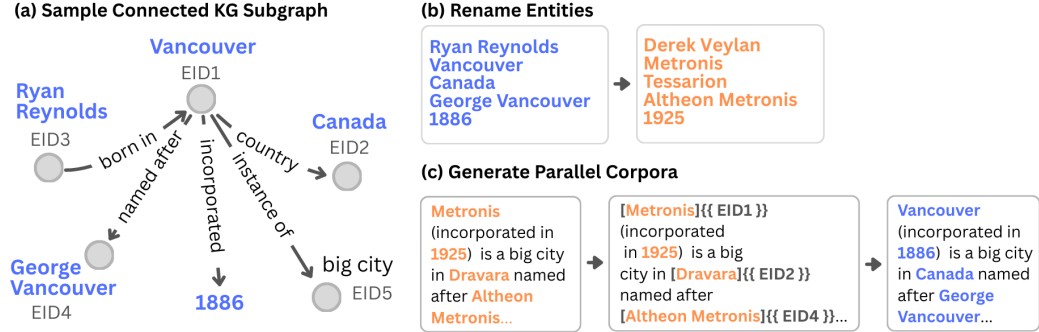

Figure 2: **Overview of SYNTHWORLDS Corpora Construction (Toy Example).** A connected subgraph is sampled from a large knowledge base (a). To obscure factual knowledge, entity labels are renamed from real-world labels (real-mapped) to synthetic name (synth-mapped) (b). From synth-mapped triplets, we generate synth-mapped documents. These documents are converted to real-mapped documents through additional LM steps with symbolic references (c). The final output is two parallel corpora: one **real-mapped**, one **synth-mapped**. Using the corpora, we construct parallel reasoning tasks (§4.1).

## 3  SYNTHWORLDS: PARALLEL CORPORA FOR CONTROLLED EVALUATION

The main idea of SYNTHWORLDS is to construct parallel corpora and tasks that describe two worlds: one grounded in real-world entities, where factual knowledge encoded in language models' parameters is potentially useful, and another built from synthetic entities, where such knowledge is deliberately uninformative. We define **factual knowledge** as entity-specific world knowledge tied to named entities (e.g., "*Barack Obama served as U.S. President from 2009 to 2017*"). In contrast, **domain-general knowledge** is not tied to named entities (e.g., arithmetic, physical laws, or the concept of an election or a university).[1] This distinction ensures that tasks maintain equivalent reasoning demands while preventing solutions that rely solely on recalling memorized entity facts. The reasoning preserved includes *commonsense* (e.g., hospitals have doctors), *compositional* (e.g., if a university has a medical school and medical schools train doctors, then the university trains doctors), *logical* (e.g., the parent of the parent of $X$ is $X$'s grandparent), and *temporal* reasoning.

**Quantifying Parametric Knowledge in Reasoning Tasks.** Constructing parallel corpora and tasks enables us to formally quantify the contribution of parametric knowledge. For a task, let $P_{\mathrm{R}}$ denote performance on the corpus with real-world entities (where parametric knowledge is useful), and $P_{\mathrm{S}}$ denote performance on the corpus with synthetic entities (where parametric knowledge is uninformative). We define the *knowledge advantage* gap as $\mathrm{KA} = P_{\mathrm{R}} - P_{\mathrm{S}}$, quantifying the contribution of parametric knowledge to task performance. We further distinguish between two settings: the baseline case, where models rely only on their parametric knowledge ($\mathrm{KA}^{\mathrm{base}} = P_{\mathrm{R}}^{\mathrm{base}} - P_{\mathrm{S}}^{\mathrm{base}}$), and the augmented case, where models are provided with external knowledge acquisition and integration strategies ($\mathrm{KA}^{\mathrm{ext}} = P_{\mathrm{R}}^{\mathrm{ext}} - P_{\mathrm{S}}^{\mathrm{ext}}$). Examples include RAG for multi-hop QA and reading page content during agentic page navigation, though agents may employ other exploration and knowledge integration strategies in novel environments. In the baseline setting, $P_{\mathrm{S}}^{\mathrm{base}}$ is expected to be near random (e.g., 0% accuracy for multi-hop QA; random walk performance for page navigation) since parametric knowledge is uninformative, so $\mathrm{KA}^{\mathrm{base}}$ reflects the pure contribution of parametric memory. Additionally, with $\mathrm{KA}^{\mathrm{base}} - \mathrm{KA}^{\mathrm{ext}}$, we quantify how much the knowledge advantage closes when allowing external knowledge integration.

**Framework Goals.** To fairly measure KA, SYNTHWORLDS corpora and tasks are constructed with four core goals: (1) **emulate real-world complexity** by capturing the structure, interconnections, and both factual consistency (facts are mutually coherent) and semantic consistency, where semantic consistency requires that surface forms remain compatible with the entity's ontological type. For example, university names remain university-like (University of Toronto → University of Grandvale, not → Grandvale Bank) and libraries remain library-like (Central Library → Oakwood Public

---

[1]Practically, we define named entities as proper nouns (i.e., capitalized) in common usage (Wikidata contributors, 2025) or recognized by NER models (e.g., the common noun *actor* vs. the named entity *Ryan Reynolds*).

Library, not → Central Stadium). This ensures surface form artifacts do not differ between real and synthetic variants, making performance on SYNTHWORLDS informative of reasoning in realistic tasks; (2) **enable parallel real- and synthetic-entity variants** to disentangle reasoning and factual knowledge; (3) **precisely control task difficulty** to support observations across levels of task complexity; and (4) **be fully automatic** such that new SYNTHWORLDS corpora and tasks can be readily constructed to continually provide novel evaluation data (guarding against evaluation corpora being included in pre- and post-training datasets).

**Obscuring Factual Knowledge in Synthetically Generated Corpora.** Similar to Wikipedia documents, SYNTHWORLDS' corpora consist of documents about a specific entity with references to other entities in the corpus. The pipeline operates in three stages (Fig 2): (1) universe construction, (2) surface-form perturbation of named entities and timestamps, and (3) document generation.

First, to ensure the world is factually consistent, the pipeline samples a universe of connected triplet facts (i.e., subject → relation → object) from an existing (and assumed to be consistent) knowledge base (Fig 2a). Next, to remove parametric knowledge while maintaining consistency, entities are systematically renamed while preserving type information and context (e.g., ensuring that the rename of *Vancouver* is still a city named after *George Vancouver* the person) (Fig 2b). Finally, based on the synth-mapped facts (using the knowledge graph structure and new synthetic names), we generate documents using LMs, following prior work on generating documents from knowledge graph facts(Fig 2c) (Agarwal et al., 2021; Josifoski et al., 2023). Specifically, we first generate documents in the synth-mapped universe consistent with the triplets. We then insert symbolic references to entities in the text. Finally, we map these references to real-mapped labels, converting each synthetic document into its real-mapped counterpart.

The pipeline outputs two parallel corpora derived from a shared set of knowledge graph triplets: one mapped to real-world entities and the other to synthetic entities. Both corpora preserve identical sentence structures and world-consistent facts, differing only in their surface-form labels. For space, we include details of SYNTHWORLDS' generation framework in Appendix A.

## 4  SYNTHWORLDS-RM AND SYNTHWORLDS-SM CORPORA AND TASKS

Using the SYNTHWORLDS framework (§3; Appendix A), we construct two parallel corpora and tasks: SYNTHWORLD-RM consisting of real-mapped entities and SYNTHWORLD-SM containing synthetic named entities. For space, we include dataset construction details in Appendix B which instantiates the framework on Wikidata. Our specific Wikidata pipeline can cheaply generate new datasets through natural stochasticity (sampling of renames) and by varying hyperparameters (e.g., entity types, seed nodes, knowledge graph sampling procedures), preventing SYNTHWORLDS from being overfit in evaluations.

| Pages | Tokens | Facts | Entity Types | Relation Types | Avg Degree | Density | # Mhop QA | # Nav Pairs |
|-------|--------|-------|--------------|----------------|------------|---------|-----------|-------------|
| 6,290 | ~1.5M | 161K | 956 | 354 | 14.6 | 0.23% | 1.2K | 1K |

Table 1: **Summary Statistics for SYNTHWORLD-RM and SYNTHWORLD-SM.**

**Dataset Statistics.** Table 1 summarizes our dataset. SYNTHWORLD-RM/SM each contain 6290 documents and over 1.5M tokens in total. The hyperlink graph is sparse, with an edge density of 0.23%. Its degree distribution is heavy-tailed: most pages have only a few links, while a small number act as hubs with disproportionately many incoming or outgoing connections. Both characteristics mirror the structure of real-world information networks such as the Web or Wikipedia (Adamic & Huberman, 2000; Kumar et al., 2000). Additional figures/tables (including cost of constructing our datasets) and qualitative examples of the dataset are provided in Appendix B.7 and B.8.

### 4.1  CASE STUDIES: PARALLEL TASKS WITH CONTROLLABLE DIFFICULTY

Given SYNTHWORLD-RM/SM corpora, we construct two tasks as case studies to evaluate LM reasoning: multi-hop QA and page navigation.

**a) Sample Subgraph with Constraints**

Ryan Reynolds — Vancouver — George Vancouver
born in → country →
Derek Veylan — Metronis — Altheon Metronis

**b) Generate Single-hop Questions**

1. Where was **Derek Veylan** born? *Metronis*

2. Who is the city of *Metronis* named after? **Altheon Metronis**

**c) Generate Multi-hop Questions**

Who is the namesake of the city where **Derek Veylan** was born? **Altheon Metronis**

**d) Map Parallel Questions**

Who is the namesake of the city where **Ryan Reynolds** was born? **George Vancouver**

Figure 3: **Multi-hop QA Construction**. Subgraphs matching reasoning motifs are sampled with constraints to ensure uniqueness, diversity, and multi-hop reasoning (a). From their triplet facts, we generate synth-mapped single-hop questions (b), which are composed into a synth-mapped multi-hop question (c). Using the synth-to-real entity mapping, we replace synth names with real names (d). The final output is parallel sets of **real-mapped** and **synth-mapped** multi-hop questions.

**Multi-hop QA.** Multi-hop questions are questions which require reasoning across multiple sources of evidence (Fig 3). For constructing these questions, we follow MuSiQue (Trivedi et al., 2022) and construct multi-hop questions through single-hop question composition (Fig. 3b). We build each multi-hop question using a specific graph motif composed of triplets, where each triplet corresponds to one single-hop question that can be composed into the final multi-hop question. This graph motif indicates a specific multi-hop reasoning structure. Table 3 summarizes all motifs used in our dataset.

Specifically, given the facts used to generate the synth-mapped documents, we first construct a global fact graph $G_{\text{facts}}$ where nodes represent entities and edges represent facts, with each fact annotated by the page where it occurs. The fact graph structure $G_{\text{facts}}$ is identical for both the synth-mapped and real-mapped corpora. From this graph, we sample subgraphs $S \subseteq G_{\text{facts}}$ that match desired reasoning motifs, ensuring that each reasoning step draws from a different page.

Next, we use an LM to generate a single-hop question for each unique triplet $(u, r, v) \in S$, where $u$ and $v$ denote the subject and object entities, respectively, and $r$ denotes the relation between them. We start with the synth-mapped entities to generate single-hop questions. For automatic quality validation, we verify that the subject entity is mentioned in the corresponding question. We prompt a LM to compose a multi-hop question from the single-hop questions. We ensure that root entities in the subgraph are mentioned in the question while all bridge entities (non-root and non-leaf) are not mentioned in the question text. Finally, to create a parallel task, we remap the entity names in both the question and answer.

This approach allows us to control task difficulty through different reasoning motifs while maintaining task parallelism by using the same sampled subgraph $S$ across both corpora. Specifically, difficulty is determined by two factors: (1) the number of hops (or equivalently, the number of decomposed single-hop questions needed to answer the question), and (2) the structural complexity of the question's motif pattern. Table 3 illustrates examples of reasoning motifs and resulting questions. In our dataset, motifs range from 2 decomposed questions (motif A) to 4 (motifs D, E, F). Motifs that contain others as subgraphs are strictly more difficult (e.g., D > B > A and F > C, E > C) since they require an additional reasoning hop. Additional details on multi-hop QA construction and ensuring task quality and diversity are included in Appendix B.4 with prompts in B.10.

**Page Navigation.** In page navigation, an agent is asked to navigate from a source to target page (e.g., navigate from *Geoffrey Hinton* to *Ryan Reynolds*) using only the hyperlinks on the page. This task is broadly related to web navigation and agentic reasoning. At each page, agents must formulate hypotheses (e.g., "the link to *University of Toronto* might lead closer to *Ryan Reynolds* since both are Canadian"), evaluate alternative decisions, and integrate information learned from prior steps (Yao et al., 2023; Wang et al., 2025). Pages that are more difficult to navigate (i.e., requiring more steps and presenting more choices at each step) further increase the demands on reasoning.

We treat the symbolic references created during document generation as hyperlinks to other pages. From this, we construct a document graph $G_{\text{doc}} = (V_{\text{doc}}, E_{\text{doc}})$ where nodes $V_{\text{doc}}$ are documents centered around specific entities and edges $(u, v) \in E_{\text{doc}}$ indicate a hyperlink from document $u$ to document $v$. Note that this graph structure is identical for both the synth-mapped and real-mapped corpora, preserving task parallelism. Creating a page navigation task simply requires specifying a source and target page. To measure and control for difficulty, we use the expected random walk distance (i.e., expected number of steps for a random walk) between two nodes as a proxy for task difficulty and sample node pairs according to different distance buckets (Chandra et al., 1989).

Higher values indicate that more intermediate decisions are required at each step, as the agent must navigate through a longer chain of choices to reach the goal.

**Task Statistics.** In total, we construct 1,200 parallel multi-hop questions spanning six reasoning structures, as well as 1,000 parallel page-navigation pairs organized into five difficulty buckets (random-walk distances of 50–1K, 1K–10K, 10K–100K, 100K–1M, and 1M–10M).

## 5 EXPERIMENTS

To study the *knowledge advantage gap*, we evaluate models on SYNTHWORLD-RM/SM, in multi-hop QA and page navigation. We evaluate six models: GPT-5-mini (OpenAI, 2025) (reasoning effort set to `medium`), Gemini-2.0-Flash (Gemini Team, 2025), gpt-oss-20B, gpt-oss-120B (OpenAI et al., 2025), Kimi-K2-Instruct (AI & the Kimi Team, 2025a), and Kimi-K2-Thinking (AI & the Kimi Team, 2025b), enabling observations across model families, model sizes, and instruct vs. thinking models. Additional experiment details and evaluation prompts are in Appendix C.

**Multi-hop QA Baselines.** We evaluate three primary baselines: (1) *Closed-book*, where the model has no access to documents and answers directly from its parametric knowledge ($KA^{base}$); (2) *One-step RAG*, where the model retrieves supporting documents once before answering ($KA^{RAG}$); and (3) *IRCoT + RAG* (Trivedi et al., 2022), which interleaves retrieval with chain-of-thought reasoning, enabling iterative reasoning and retrieval steps($KA^{CoT + RAG}$). For retrieval, we use the HippoRAG 2 retriever, designed for factual, multi-hop contexts (Gutierrez et al., 2024). In addition, we include a *Reading Comprehension* condition in which the model is given all gold (2-4 documents depending on graph motif, examples in Table 3) and additional distractor documents, equaling 10 total. This condition serves two interpretations: (i) it provides an upper bound when retrieval is not a bottleneck, and (ii) it separates the inherent difficulty of the reasoning task from the challenge of retrieving relevant evidence in unfamiliar settings. All baseline prompts for QA are included in Appendix C.1.

**Page Navigation Baselines.** Page navigation tests an agent's ability to plan and reason over a linked knowledge environment. For page navigation, we follow the design of existing tool-use agents (Yang et al., 2024; Gu et al., 2025) and evaluate an agent equipped with two function-calling tools: `click_link`, which allows the agent to click any link on the current page, and `backtrack`, which allows the agent to return to a previously visited page. To address our navigation research questions, we evaluate the agent under two observation conditions: (1) *Links Only*, where the agent observes only the set of outgoing links on each page ($KA^{base}$); and (2) *Content + Links*, where the agent observes both the outgoing links and the full page text ($KA^{content}$). We include all prompts for agentic navigation in Appendix C.2.

The *Links Only* condition isolates the contribution of parametric knowledge and semantic familiarity, since navigation must rely entirely on recognizing entities in link text. The *Content + Links* condition tests whether access to textual content can compensate for the absence of parametric knowledge by providing additional evidence for navigation decisions. In both settings, the agent is limited to a maximum of 30 steps. This cap is well above the distribution of shortest path lengths (median 5, maximum 11), ensuring all tasks remain solvable while avoiding unbounded exploration. In our subsequent results, we observe that this bound is sufficient for meaningful exploration.

**Metrics.** For all multi-hop QA experiments, we report token-based F1 scores for task performance following prior work (Trivedi et al., 2022). Following HippoRag 2 (Gutiérrez et al., 2025), we also report recall@5 for RAG baselines to evaluate retrieval quality. For page navigation, we report the success rate of reaching the target page.

## 6 RESULTS AND DISCUSSION

We show results across task buckets for multi-hop QA and page navigation in Figures 4 and 5. We report aggregated results for all task instances in Table 4 and 5 in the Appendix.

**RQ1: What is the knowledge advantage gap when relying solely on parametric knowledge?**
In multi-hop QA, across models, we observe the baseline performance in RM, $P_R^{base} \approx 20$, indicating that SYNTHWORLD-RM presents questions that LMs *can* answer using parametric knowledge

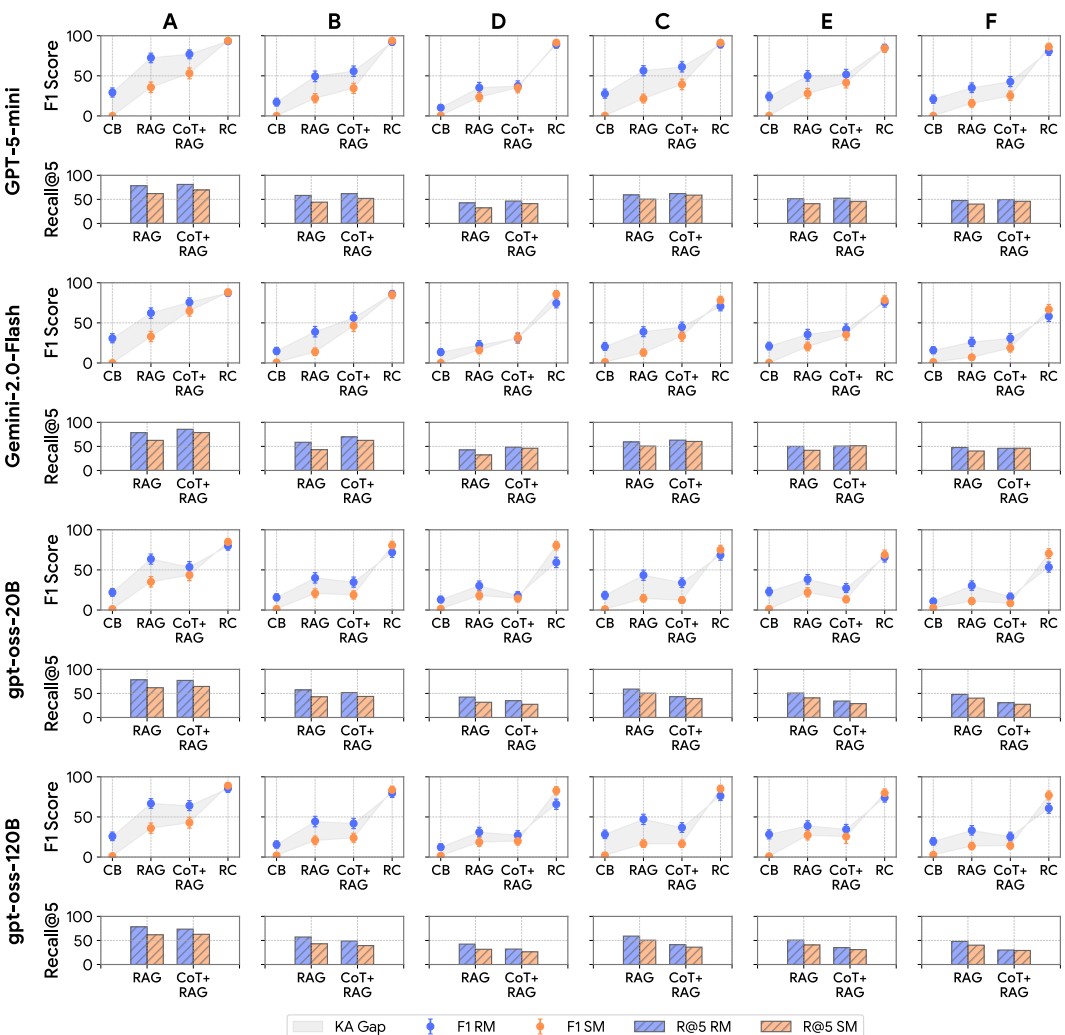

Figure 4: **Multi-hop QA Results by Reasoning Motifs**. We report F1 scores on SYNTHWORLD-RM (RM) and SYNTHWORLD-SM (SM), along with the *knowledge advantage gap* (KA = $F1_{RM} - F1_{SM}$). Settings: CB = Closed-book, RAG = One-step RAG, CoT+RAG = IRCoT + RAG, RC = Reading Comprehension. We show Recall@5 for RAG baselines (by construction, CB has recall = 0 and RC has recall = 1). IRCoT + RAG substantially reduces the KA gap compared to the CB baseline, primarily due to improved retrieval. Example questions for each motif are given in Table 3.

(Table 4; Closed-book, RM). In contrast, the near-zero $P_S^{base}$ validates that SYNTHWORLD-SM questions *cannot* be solved with parametric knowledge alone (Table 4; Closed-book, SM). Overall, $KA^{base} \approx 20$ (Table 4; Closed-book, KA). As task difficulty increases, $P_R^{base}$ decreases as expected, while $P_S^{base}$ remains at 0, showing that the gap would be even wider if we restricted evaluation to easier QA tasks (Fig. 4; CB left to right). In the reading comprehension setting, performance is equalized or even stronger in the SM cases because LMs are not distracted by parametric knowledge that could interfere with grounding its reasoning in the content (Monea et al., 2024).

For page navigation, we find a larger gap for GPT-5-mini and Kimi-K2 models ($KA^{base} \approx 30$) than for Gemini-2.0-Flashand gpt-oss models ($KA^{base} \approx 20$) (Table 5; Links Only, KA). This suggests the first set of models are better able to leverage parametric knowledge to locate the target page. Across difficulty levels, performance drops for both RM and SM tasks, but the gap persists. At the easiest difficulty, the gap narrows slightly, as models in SM can exploit the structure and semantics of hyperlinks to achieve modest success.

**RQ2: To what extent does knowledge augmentation help close the gap?** Knowledge augmentation with One-step RAG improves absolute performance across both RM and SM tasks. However,

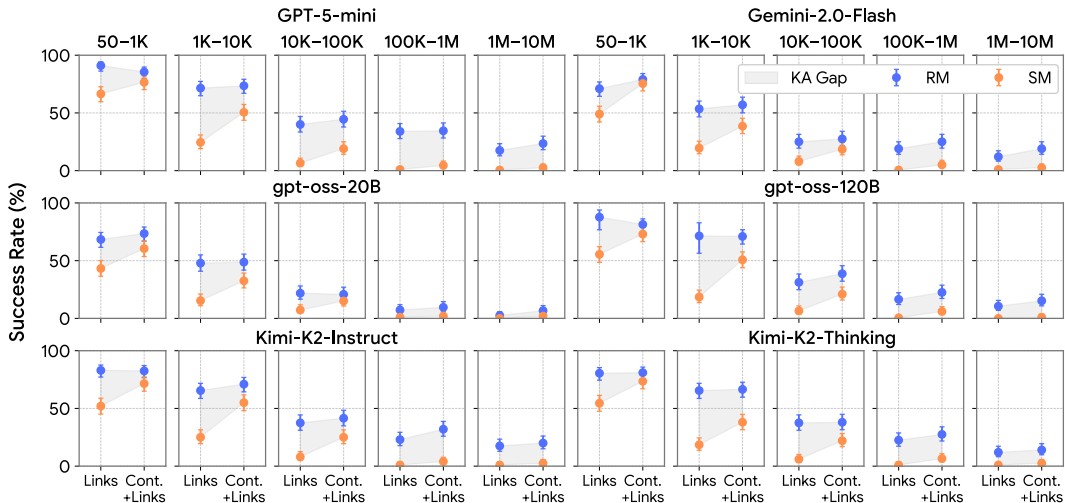

Figure 5: **Page Navigation Results by Difficulty (i.e., Expected Random Walk Distance)**. We report success rate on SYNTHWORLD-RM (RM) and SYNTHWORLD-SM (SM) and the *knowledge advantage gap* (KA $=$ Success$_{RM}$ $-$ Success$_{SM}$). Models consistently perform better on real-mapped corpora, especially in harder navigation tasks, indicating that parametric knowledge enables shortcuts. Page content (*Content + Links* vs. *Links Only*) benefits models more on synth-mapped corpora, narrowing the gap and showing its value in novel environments.

the knowledge advantage does not shrink; in fact, it widens. Specifically, $KA^{base} - KA^{RAG} = -4.0$ for GPT-5-mini and $-1.3$ for Gemini-2.0-Flash and similarly for other models (Table 4; Closed-book – One-step RAG), a pattern consistent across multiple difficulty levels (Fig. 4; A, B, C, F). This suggests that while One-step RAG benefits both RM and SM, it disproportionately benefits RM and reinforces models' reliance on parametric knowledge. Meanwhile, IRCoT + RAG reduces the gap. Overall, $KA^{base} - KA^{IRCoT+RAG}$ is positive for both models, 5.2 for GPT-5-mini and 10.3 for Gemini 2.0-Flash (Table 4; Closed-book – IRCoT + RAG). We observe the gap closing across reasoning motifs (Fig. 4), indicating that interleaving retrieval with reasoning better aligns knowledge integration with task demands. [2]

To further probe this effect, we compare with the reading comprehension setting (i.e., perfect recall by construction). Triangulating reading comprehension F1-scores with F1-scores and retrieval recall from One-step RAG and IRCoT + RAG (Fig. 4; rows 2 and 4), we can infer that retrieval quality is a main driver of observed performance gaps. Retrieval performance improves slightly with IRCoT in both RM and SM, but retrieval in SM remains consistently lower than in RM. HippoRAG 2 uses an LM (GPT-5-mini in our experiments) to separately index RM and SM corpora and to retrieve documents given the input query. Given this setup, our results suggest that LM-based retrievers may not generalize well in novel environments, raising questions about the robustness of LM-indexed retrieval pipelines.

With respect to page navigation, across all task instance, we observe granting the agent access to page content improves performance, yielding differences of $KA^{base} - KA^{content} = 9.3$ and 7.0 for GPT-5-mini and Gemini-2.0-Flash, respectively (Table 5; Links Only – Content + Links). The performance gap narrows most on simpler navigation pairs (Fig. 5), though it remains present on more difficult ones.

To potentially explain the knowledge advantage gap, we analyze agent behavior by measuring how often externalized reasoning traces mention entities not observed during page navigation. For example, when tasked with navigating to the Brussels metropolitan area, a model trace included the statement: *"Ghent is in Belgium and likely links to Belgian geography or Brussels-related pages."* We count the mentions of *Belgium* and *Belgian* as external, since they had not appeared in any previously visited page. In the SM setting, this rate is 0 by construction (and confirmed empirically).

---

[2] We note IRCoT + RAG does not improve absolute performance compared to One-step RAG for gpt-oss models as they struggle to follow the IRCoT prompt format. These results point to the importance of nuanced studies into the impact of knowledge integration.

Meanwhile, in the RM setting, we observe frequent reliance on external knowledge: under the *Links Only* condition, at least one external entity is mentioned in 48% of steps for GPT-5-mini and 60% for Gemini-2.0-Flash. Expanding access to *Content + Links* reduces these rates to 35% and 15%, respectively. Without page content, RM models tend to fall back on stored factual knowledge. In contrast, SM-like settings (where information is novel) offer only limited scope for fallback. This points to an opportunity to design agentic systems that both remain effective and efficiently acquire the necessary background knowledge.

**Insights enabled by SYNTHWORLDS.** The parallelism of SYNTHWORLDS enables controlled comparisons that isolate different aspects of model behavior. For example, it can allow us to ask when models take longer reasoning paths in the absence of recall or whether (and under what conditions) error types shift. It also makes it possible to investigate which system-level factors (such as retrieval quality in QA) and which core LM capabilities (as measured by reasoning or agentic benchmarks) lead to narrower or wider knowledge advantage gaps. In our experiments, we studied knowledge integration through retrieval, both in single-step RAG and when interleaved with chain-of-thought or agentic workflows. These methods improved performance but did not fully eliminate the knowledge advantage gap. In QA, we see that it is a problem about knowledge acquisition (i.e., obtaining all the relevant documents), but additional thinking (e.g., CoT) can help. Meanwhile, in page navigation, even when models have the same content available, there is a gap as factual knowledge enables shortcuts. Beyond our case study results, SYNTHWORLDS allows researchers to examine alternative integration schemes. For example, in page navigation, what if models are integrated with retrieval to better plan their navigation? To what extent do long-context methods, where models must synthesize and retain relevant information without retrieval (Hsieh et al., 2024), or multi-agent workflows (Du et al., 2024), where group discussion and feedback shape integration, can help with knowledge augmentation?

**Future work and extending SYNTHWORLDS.** Our current work only scratches the surface of these possibilities. A limitation is that our experiments were conducted on the specific SYNTHWORLDS corpora and task designs we introduced, which may restrict the generality of our findings. These choices do not cover the full space of "constructed worlds" (or tasks) that could be defined by different relation types, connective structures, or contexts. Altering the way the corpora is constructed could lead to different outcomes. Nonetheless, because SYNTHWORLDS is fully automatic, inexpensive, and flexible given any input knowledge base, we can generate alternate parallel corpora and probe these questions more broadly (see Appendix B.6 for an expanded discussion). Future work could impose targeted constraints on graph construction to highlight particular reasoning challenges or examine how parametric knowledge interacts with different underlying knowledge bases.

In general, our framework requires a high-quality knowledge graph to encode complex relationships and ensure factual consistency in synthesized text. Therefore, as modern extraction methods make knowledge graph construction from text increasingly feasible (Sainz et al., 2024; Xu et al., 2024a), graphs can be constructed directly from unstructured text (e.g., Wikipedia). Once built with consistent, accurate facts, the SYNTHWORLDS framework follows naturally. Likewise, SYNTHWORLDS is not limited to one knowledge graph or domain. The core idea, i.e., sampling a subgraph, renaming entities, and constructing parallel corpora and tasks, generalizes to any knowledge graph. The main requirement is understanding which entities should be renamed, which depends on the domain. For example, in mathematics, we could create parallel worlds with different notation systems (RM: $x, y, f(x)$; SM: $\alpha, \beta, \phi(\alpha)$). Similarly, for code generation, we can consistently rename entire libraries (e.g., `numpy`/`pandas`) and function calls for the SM variant. By supporting controlled studies of reasoning, memory, and adaptation across varied settings, SYNTHWORLDS lays the groundwork for developing LM systems that are more robust and generalizable.

## 7 CONCLUSION

We present SYNTHWORLDS, a framework for disentangling the role of parametric knowledge in LM reasoning and retrieval. By constructing parallel corpora and tasks with controllable difficulty, SYNTHWORLDS reveals persistent performance gaps even when models have access to retrieval or page content. These findings highlight opportunities for advancing reasoning in novel environments and position SYNTHWORLDS as a scalable testbed for developing methods that generalize beyond reliance on parametric knowledge.

## 8  REPRODUCIBILITY

We provide full details of dataset construction, experimental setup, hyperparameters, and prompts in the Appendix ensuring that our dataset and results could be reproduced. The dataset used in our experiments is included in the supplementary material and will be publicly released. The code and dataset for running all experiments is available at `https://anonymous.4open.science/r/synthworld-experiments-CE26/`.

### ACKNOWLEDGMENTS

We thank the UW Behavioral Data Science Group members, Jeffrey Li, Weijia Shi, and Harsh Trivedi for their valuable suggestions and feedback, and Tiffany Zheng for her continued motivation and personal support throughout this project. This research was supported in part by NSF IIS-1901386, NSF CAREER IIS-2142794, and a Garvey Institute Innovation grant.

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

APPENDIX TABLE OF CONTENTS

# A  SYNTHWORLDS FRAMEWORK

In this section, we discuss the core formalization of the SYNTHWORLDS framework. Concrete details actualizing this framework in our SYNTHWORLD-RM/SM datasets are included in Appendix B.

**World Knowledge Preliminaries.**  Formally, our dataset generation takes as input a knowledge base KG consisting of a collection of entities $\mathcal{E}$ and a collection of relations $\mathcal{R}$. We define the set of facts as $\mathcal{F} \subseteq \mathcal{E} \times \mathcal{R} \times \mathcal{E}$, and represent the corresponding graph as $G = (\mathcal{E}, \mathcal{F})$. Each entity $e \in \mathcal{E}$ has an associated label $\ell(e) \in \mathcal{L}$, where $\mathcal{L}$ denotes the space of surface-form names (e.g., textual strings such as "Albert Einstein"). In addition, each entity includes a relation of the form $(e, \mathrm{ent\_type}, \tau(e))$, where $\tau(e) \in \mathcal{T}$ specifies the entity's ontological type (e.g., person, house, plane). $\tau(e)$ is intended to denote a general category, without mention of specific named entities.

A universe of triplet facts is therefore defined by $U = (G, \ell)$.

**Coherent Universe Construction.**  To construct a coherent and connected universe we leverage the facts from $G$. At a desired tractable size and complexity, we first sample a connected subgraph $G' \subseteq G$ (Fig. 2a). $G'$ is constructed by iteratively expanding the frontier from a seed set $\mathcal{Q}_0 \subseteq \mathcal{E}$. At iteration $t$, given the current frontier $\mathcal{Q}_t \subseteq \mathcal{E}$, we sample neighbors $\mathcal{N}(v)$ for each $v \in \mathcal{Q}_t$ and add them to the subgraph. Here, $\mathcal{N}(v)$ includes all entities $u$ such that $(v, r, u) \in \mathcal{F}$ or $(u, r, v) \in \mathcal{F}$ for some $r \in \mathcal{R}$.

After $T$ expansion steps we obtain a sampled subgraph $G_T \subseteq G$. To ensure sufficient connectivity, we extract the $k$-core subgraph (i.e., the maximal subgraph in which every node has degree at least $k$) and then take its largest connected component, denoted $G_{T,k} \subseteq G_T$. For notational simplicity, in the following we use $G$ to refer to $G_{T,k}$.

**Surface-Form Perturbations.**  To obscure factual knowledge, we perturb surface forms, i.e., entity names and timestamps tied to entities (Fig. 2b).[3]

Simple renaming risks (a) **factual leakage**, where replacements still reveal real-world associations (e.g., *Tokyo → Torioka*, which continues to suggest Japanese origins)), or (b) **incoherence**, where substitutions violate type or consistency constraints (e.g., *Ryan Reynolds was born in Vancouver → Silvercrest Collegiate was born in Sarah Thompson*), thereby failing to preserve domain-general knowledge. To prevent these issues, we systematically perturb all named entities and temporal labels through controlled renaming that obscures underlying facts while preserving coherence.

In particular, this entails: (i) **type-consistent naming**, where synthetic names respect the entity's ontological type (e.g., *Nile River → Lora River*, not *Lora Pavilion*), and (ii) **name-derivation consistency**, where renames propagate to related surface forms (e.g., if *Vancouver → Metronis*, then *George Vancouver*, after whom the city is named, *→ Altheon Metronis*). These constraints preserve semantic coherence and affiliation cues, preventing surface-level artifacts from confounding evaluation.

Let $\mathcal{E}_{\mathrm{proper}} \subseteq \mathcal{E}$ denote the set of named-entity nodes subject to renaming and $\mathcal{L}_{\mathrm{real}}$ the denote the set of original real-mapped labels. We say that node $u$ is *name-related* to node $v$ if and only if $u, v \in \mathcal{E}_{\mathrm{proper}}$ and (i) $\ell(v)$ is a substring of $\ell(u)$ and (ii) $\exists r \in \mathcal{R} : (u, r, v) \in \mathcal{F}$ or $(v, r, u) \in \mathcal{F}$. That is, name-relation requires both a lexical substring relationship and an explicit relation in the knowledge graph. For instance, *Vancouver* is name-related to *Vancouver Canucks*.

This induces a directed acyclic *name-related dependency graph* $G_{\mathrm{dep}} = (\mathcal{E}_{\mathrm{proper}}, E_{\mathrm{dep}})$ where $(u, r, v) \in E_{\mathrm{dep}}$ if and only if $u$ is name-related to $v$ with relation $r$. We rename entities according to a level-order (breadth-first) traversal of $G_{\mathrm{dep}}$, processing all nodes at each level before moving to the next level. This ensures that all entities at depth $d$ are newly labeled before any entity at depth $d + 1$, maintaining consistency across substring relationships.

We define the updated labeling function $\ell' : \mathcal{E} \to \mathcal{L}$ through the following process. For each $v \in \mathcal{E}_{\mathrm{proper}}$ processed in level-order, we query a LM with input $(\tau(v), \{(\ell'(u), \tau(u), r) : (u, r, v) \in E_{\mathrm{dep}}\})$ to generate $\ell'(v)$. In other words, we rename entities by providing the LM with the target

---

[3]Other literals, e.g., population counts and physical measurements, are excluded because they could easily (a) reveal real-world facts (e.g., "*Mount FakeMountain is 8848m tall*" still points to Mount Everest) or (b) distort domain-general reasoning when perturbed.

entity's type and the new names of all related entities it depends on. For entities not being renamed, we set $\ell'(e) = \ell(e)$. We include prompts for renaming in Appendix B.9.

For timestamps, we apply a fixed offset $\delta$ per universe: for any timestamp $x$, we replace it with $x+\delta$, preserving ordering and interval relations (e.g., a parent's birth precedes a child's), while removing the potential for parametric knowledge to be leaked.

After these perturbations, we produce a synth-mapped universe $U' = (G, \ell')$ where entities retain their structure and types but receive new synthetic labels.

**Parallel Corpora Generation.** For corpora generation (Fig. 2c), we first generate documents from the synth-mapped universe $U'$ such that the facts are faithful to $G$, then add symbolic references to entity IDs in the text, before using these IDs references mapped to real-mapped labels to covert each synthetic document into a real-mapped version. The output is two parallel corpora: one synth-mapped and one real-mapped with identical sentence structures and world-consistent facts, differing only in their surface-form labels.

By generating documents from synth-mapped (as opposed to real-mapped) entities first, we exploit the asymmetry that synthetic entity names $\ell'(e)$ have no connections to the LM's parametric knowledge. This prevents the LM from introducing auxiliary facts and makes it easier to stay faithful to the provided triplets. For example, when writing about the synthetically named entity for *Austria*, the LM cannot mention facts about *Vienna* based on external knowledge and must rely solely on the provided facts.

Concretely, for each entity $v \in \mathcal{E}$, we collect all incident edges $\{(u,r,v) \mid (u,r,v) \in \mathcal{F}\} \cup \{(v,r,u) \mid (v,r,u) \in \mathcal{F}\}$ and retain only the majority orientation (i.e., whichever set is larger) to define $N(v)$. We then query an LM to generate a document describing the facts in $N(v)$.[4]

Next, following Hennigen et al. (2024), we instruct an LM to add symbolic references $\{e_1, e_2, \ldots\}$ to the synth-mapped documents, adding to each mention of $\ell'(e)$ a symbolic identifier. This provides both hyperlinks for document navigation (§4.1) and facilitates the conversion process described.

Given a synthetic document with symbolic references and the entity mapping $\{(e, \ell(e), \ell'(e)) : e \in \mathcal{E}\}$, we query an LM to generate an equivalent real-mapped document by replacing each symbolic reference $e_i$ with the original label $\ell(e_i)$. The symbolic references ensure that the correct entity mapping is preserved during conversion. During this process, we apply programmatic and LM-based checks to ensure document parallelism, factual consistency, and effective knowledge obfuscation.

## B SYNTHWORLD-RM/SM DATASET CONSTRUCTION DETAILS

Our dataset construction pipeline follows the framework in Appendix A (overview in Fig. 2). All prompts for dataset construction are in Appendix B.9- B.10. Table 2 summarizes the LM used and LM API costs for each step of the pipeline including multi-hop QA task construction.

### B.1 UNIVERSE CONSTRUCTION

For our specific SYNTHWORLDS corpora we start with the Wikidata KG (Vrandečić, 2012) (01/20/2025 dump).

Knowledge graphs such as Wikidata are heavily skewed toward a small set of high-frequency relations (e.g., instance of, subclass of, located in). If we sample subgraphs in strict proportion to this distribution, the resulting universe is both narrow in structure and closely aligned with the original world knowledge. This limits its usefulness for tasks where we want to probe reasoning in settings that are not simply memorization of facts. To control edge-type diversity, we introduce a *uniformity factor*. For $v \in \mathcal{E}$ at iteration $t$, let $\Gamma_t(r; v)$ denote the set of candidate triplets involving $v$ with relation $r$. We define

$$P_t(r \mid v) = \frac{|\Gamma_t(r; v)|^\alpha}{\sum_k |\Gamma_t(k; v)|^\alpha}, \quad \alpha = 1 - \text{uniformity}.$$

---

[4]In initial experiments, including both orientations often led the LM to generate inconsistent documents, e.g., an entity described as both the son and the father of another.

High uniformity yields diverse edge types ($\alpha = 0$: uniform), while low uniformity favors frequent relations ($\alpha = 1$: frequency-proportional).

To encourage diversity of entities, we initialize $\mathcal{Q}_0$ as the set of Wikidata entities across all categories defined in Wikipedia's popular pages (contributors, 2025) To ensure high-quality entities, we discard Wikidata nodes that are time terms, Wikimedia-bookkeeping entities, unlabeled entries, or entities whose names include numbers. We run the iterative sampling for $T = 11$ steps with uniformity $= 0.6$, and take the 19-core subgraph $G' = G_{11,19}$.

## B.2    SURFACE-FORM PERTURBATIONS

We rename entities identified via Wikidata's entity naming rules.[5]

Given all proper-name entities $\mathcal{E}'_{proper}$ in $G'$ that share a type description, we prompt a LM to propose new names for that entity type following. In Wikidata, entitiy type is inferred through the instance of relationship (P31). However, certain instance of continue to contain named entities. For these cases we recursively apply the instance of until no named entities exist in the label. For example, say Vancouver only has a instance of label "city in British Columbia" in this case we take the instance of label for British Columbia which is "province of Canada", finally we take the label for Canada which is country so then the label becomes "city in province of country".

In addition, we incorporate Wikidata time qualifiers[6] (e.g., Barack Obama $\rightarrow$ president $\rightarrow$ USA; start time $\rightarrow$ 20 January 2009), which attach additional temporal information to fact triplets. To prevent timestamps from trivially revealing real-world identities, we apply a $\delta = 39$.

## B.3    PARALLEL DOCUMENT GENERATION

We prompt a LM to generate a factually consistent document from fact triplets (prompts in Appendix B.9). To ensure quality, we add the Wikidata entity id (prefixed with Q, e.g., Q15) when generating symbolic references. These are unique identifiers for the underlying entity that we can then use to check the correct label is used in the corresponding real-mapped and synth-mapped documents. We implement programmatic checks to guarantee that (1) only entities present in the facts are included in the page, and (2) the display text for each entity matches the underlying link. When converting from synth-mapped to real-mapped text, we additionally require that both documents share the same set of symbolic references (thus inducing the same graph structure) and that no mention of any synth-mapped entity remains. Finally, we enforce strict quality thresholds: we only keep pages when (a) the similarity (measured using the Damerau-Levenshtein edit distance (Damerau, 1964)) between the initial generation and the symbolic-reference version exceeds 0.95, and (b) the similarity between the synth-mapped and real-mapped versions exceeds with symbolic references exceeds 0.85. Practically, this filtering ensures that only parallel documents with highly consistent structure and minimal unintended variation are retained.

To ensure that the generated pages are truly novel, we prompt the same LM to guess the underlying entity from a synth-mapped document, providing it with the (unrealistic) clue that the page corresponds to a real-mapped entity whose names have been perturbed. This constitutes a deliberately strict check: in actual task settings, the LM would never be told that the page is based on a real-world entity. Any page the LM gets correct we remove from our corpus. After each filtering step, we retain only the largest connected component of the hyperlink graph, ensuring that the resulting corpus remains navigable for downstream page-navigation tasks.

## B.4    MULTI-HOP QA CONSTRUCTION

**Validating Facts for QA Construction.** Prior to the steps described in Section 4.1, we also first validated what facts were actually in the generated corpora. This step accounts for cases where some facts may have been omitted during generation. Given a document generated by a LM and the set of source facts the generation based on, we use another LM to identify which of those facts are actually present in the document. The prompt for this step is included in Appendix B.10.

---

[5]https://www.wikidata.org/wiki/Help:Label
[6]https://www.wikidata.org/wiki/Help:Qualifiers

This step enables us to construct the directed fact graph $G_{\text{fact}} = (\mathcal{E}, \mathcal{F})$. Each fact is a directed triple

$$(e_i, r, e_j) \in \mathcal{F}, \quad e_i, e_j \in \mathcal{E},$$

where $r$ is a relation annotated with a property name, and the source page in our corpora from which the fact was extracted. By construction, each edge originates from a distinct source page, ensuring that multi-edge subgraphs aggregate knowledge across independent contexts.

**Ensuring Diversity of Generated Questions.** Given the fact graph we sample graph motifs (i.e., the motifs in Table 3). A *motif* is a relational subgraph of $G_{\text{fact}}$, defined as

$$\mathcal{M} = (\mathcal{V}_M, \mathcal{F}_M), \quad \mathcal{V}_M \subseteq \mathcal{E}, \ \mathcal{F}_M \subseteq \mathcal{F}.$$

To ensure diversity and quality of questions generated, we sample graphs subject to the following constraints:

1. All entities in a motif must be distinct: $e_i \neq e_j \quad \forall i \neq j, \ e_i, e_j \in \mathcal{V}_M$.

2. All facts in $\mathcal{F}_M$ must come from different pages.

3. For a given anchor configuration and relation sequence, at most one instantiation of the motif is retained. For example, for motif A, we keep at most one subgraph $\{(e_1, r_1, e_2), (e_2, r_2, e_3)\}$, for each tuple $(e_1, r_1, r_2)$. For motif E, we keep at most one subgraph $\{(e_1, r_1, e_2), (e_3, r_2, e_4), (e_2, r_3, e_5), (e_4, r_4, e_5), (e_5, r_5, e_6)\}$, for each tuple $(e_1, e_3, r_1, r_2, r_3, r_4, r_5)$. In other words, we ensure there is only one unique **reasoning chain** for a given motif.

4. Following Trivedi et al. (2022), we remove any n-hop question that is a sub-graph of any m-hop question (m > n > 1).

5. To prevent over-representation of any particular edge or intermediate node, we limit reuse of facts and bridge entities within motifs. Concretely, each fact $(e_i, r, e_j) \in \mathcal{F}$ and each bridge entity (i.e., entities that are neither roots nor terminal nodes of a motif) is sampled at most five times per motif.

## B.5 HUMAN VALIDATION

To assess corpora quality, two researchers labeled each candidate fact as (i) *expressed in the document*, (ii) *not expressed*, or (iii) *inconsistent with the document*. Across 28 unique pages ($n = 798$ facts), no inconsistencies were observed, giving a 95% upper bound of $0.4\%$ on the true inconsistency rate. On 7 double-annotated pages, agreement was 99.5% with Cohen's $\kappa = 0.85$, indicating almost perfect reliability. Corpus-level factual recall was 98.8% (95% CI [98.0, 99.7]), with mean page recall 98.9%. These results demonstrate that the dataset is clean, reliable, and faithfully represents the intended facts.

To validate question quality, another researcher inspected a sample of 30 parallel questions, covering 5 examples for each reasoning motif. For each question, the researcher verified three criteria: (i) the questions were parallel (ii) the question led to a correct and unambiguous answer, and (iii) the resulting question was coherent and natural. All questions were found satisfactory.

## B.6 DISCUSSION ON DATASET CONSTRUCTION

**Choice of Entities to Rename.** During corpus construction, we restrict renaming to Wikidata entities whose labels begin with a capital letter (e.g., *Geoffrey Hinton*, Q92894), which typically indicates named entities. Entities whose labels begin with lowercase letters (e.g., *dog*, Q144; *oxygen*, Q629) are not renamed. An edge case arises for entities such as *einsteinium* (Q1103), the element named after Albert Einstein. Since *einsteinium* does not begin with a capital letter, it would not be renamed, creating a potential factual knowledge leak (e.g., "*einsteinium* is named after [Renamed Scientist]" implicitly revealing *Albert Einstein*). To mitigate this, we remove all synth-mapped pages where such leakage could occur, ensuring that models cannot trivially recover world knowledge after being told that entities have been renamed. Obfuscating *einsteinium*-style knowledge more broadly and directly remains an avenue for future work.

**Controllability and Stochasticity in Data Generation.** To generate new instances of SYNTH-WORLDS, we expose several controllable knobs. Different seed nodes (e.g., starting with AI researchers) can be sampled to produce distinct yet structurally valid corpora. The uniformity factor can be varied to influence graph connectivity. Subgraph sampling can also be restricted to entities of specific types (e.g., researchers, institutions, students), or emphasize/de-emphasize particular edge relations. Renaming strategies further contribute variability: alternative LMs, different temperature settings, or varied timestamp perturbations can all yield distinct datasets. Finally, document generation may use different LMs to produce stylistic variation, while remaining consistent with the underlying facts. Together, these controls balance the need for world consistency with stochastic diversity across dataset instantiations.

## B.7 ADDITIONAL FIGURES AND TABLES

Figure 6 shows the distributions of page entity types (based on Wikidata's `instance of` property) and relation types (across all facts) in the generated corpora. Figure 7 shows the in-degree and out-degree distributions of the page graph in SYNTHWORLDS. Figure 8 visualizes the constructed hyperlink graph used for Page Navigation. Table 2 provides the LM API cost of constructing SYN-THWORLD-RM/SM. Table 3 includes all graph motifs and examples of constructed questions.

| Dataset Construction Step | LM Used | API Calls | Inp Tok | Out Tok | Cost ($) |
|---|---|---|---|---|---|
| Surface Form Renaming | GPT-4o-mini | 0.3K | 237.9K | 38.8K | $0.06 |
| Corpora Generation | GPT-5-mini | 35.3K | 110.1M | 74.3M | $176 |
| Novelty Validation | GPT-5-mini | 15.5K | 6.7M | 93.4M | $188 |
| Multihop-QA Question Gen. | GPT-5-mini | 4.8K | 6.9M | 3.0M | $7.82 |
| **Total** | — | **55.9K** | **123.9M** | **170.7M** | **$372** |

Table 2: **Token usage and LM API costs for constructing SYNTHWORLD-RM/SM.** Totals are shown in the last row. During the project period new LMs were released and we sought to use the best models available to generate a public datasets. This means that GPT-4o-mini was used during surface form renaming (a much simpler task) while all other steps used GPT-5-mini. The number of API calls includes follow-up prompts when the initial LM output does not pass programmatic validation checks. GPT-5-mini was used with the default reasoning effort set to `medium`. For novelty validation, we enforced a very strict notion of novelty and explicitly instructed the model to "think", which inflated reasoning token usage (details in §B.3; prompt in §B.9). In practice, one could reduce reasoning effort to `low`, since faithful evaluation on synth-mapped tasks would not prompt LMs with the information that entities have been renamed. Such adjustments would substantially lower costs, bringing the total closer to $200.

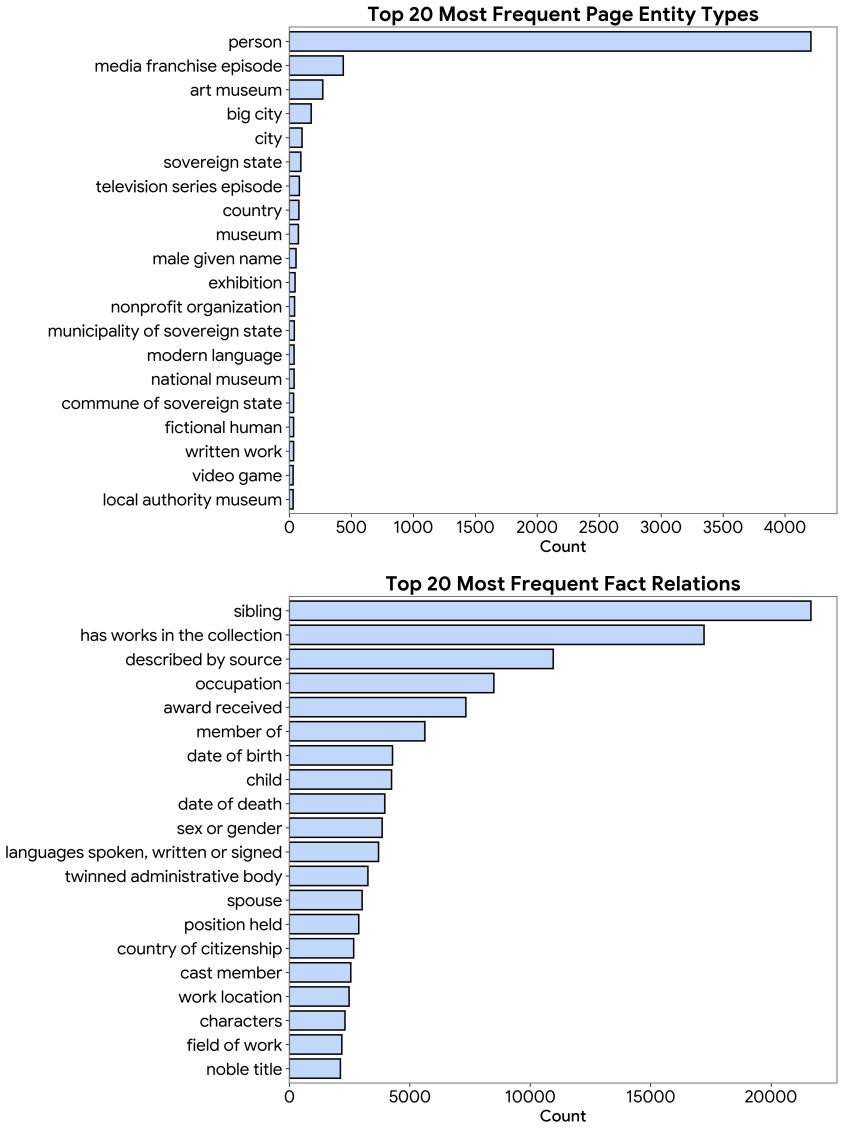

Figure 6: **Entity Type and Relation Type Distribution of SYNTHWORLD-RM/SM.** Documents cover a broad range of entity types and relation types.

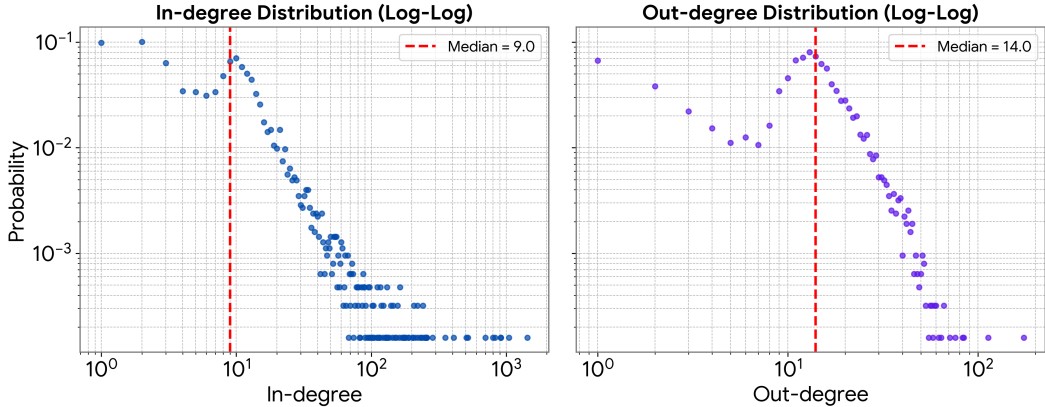

Figure 7: **Degree Distribution of SYNTHWORLD-RM/SM.** Our corpora preserve the interconnected and structured nature of knowledge networks (i.e., power-law degree distribution), matching the complexity of real-world information ecosystems.

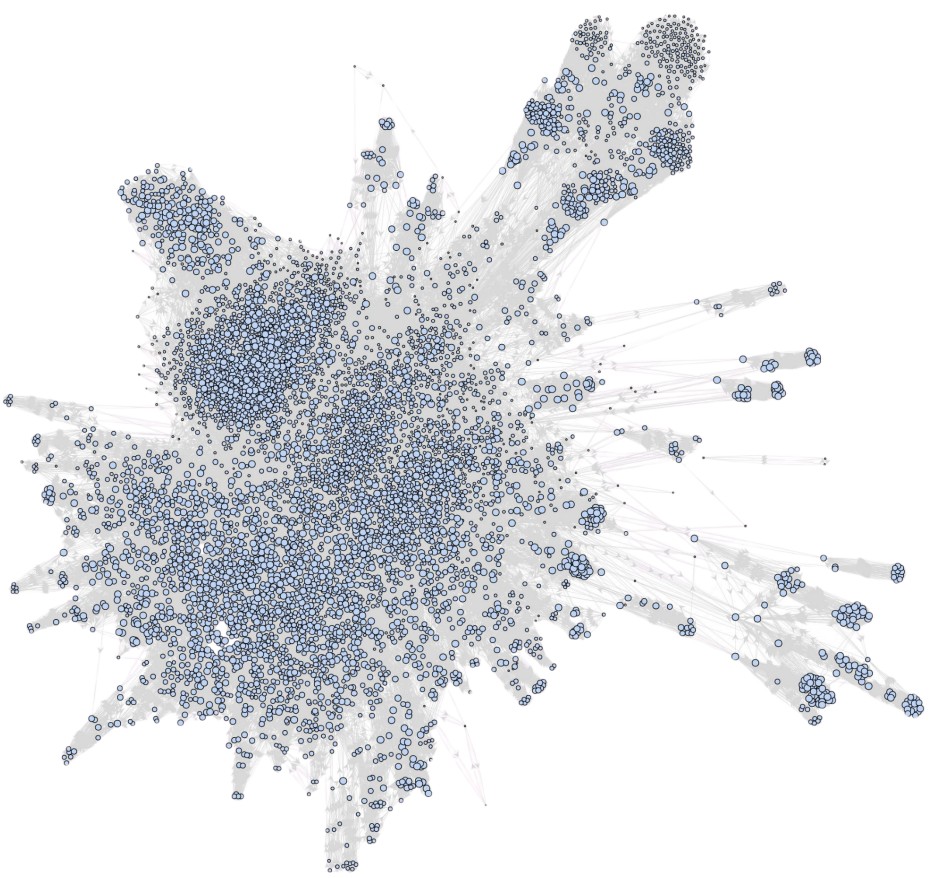

Figure 8: **SYNTHWORLD-RM/SM Hyperlink Graph** illustrating a scale-free topology, where a few highly connected hubs dominate while most nodes have relatively few links. Node size is determined by $\max(1, \min(4, \frac{\deg(v)}{8}))$.

| Graph | Motif | Decomposition | Question |
|---|---|---|---|
| A | | **1.** Who was the screenwriter of **The City on the Edge of Forever**? *Harlan Ellison* **2.** In what year was **Harlan Ellison** nominated for **Hugo Award for Best Short Story**? *1971* | In what year was the screenwriter of **The City on the Edge of Forever** nominated for **Hugo Award for Best Short Story**? *1971* |
| B | | **1.** Which family does **Sirindhorn, Princess Royal** belong to? *House of Mahidol* **2.** Who is the chairperson of **House of Mahidol**? *Vajiralongkorn* **3.** Where does **Vajiralongkorn** live? *Grand Palace* | Where does the chairperson of **Sirindhorn, Princess Royal**'s family live? *Grand Palace* |
| C | | **1.** Who is **Johann Bernoulli**'s doctoral student? *Daniel Bernoulli* **2.** Who was Alexander R. Todd, Baron Todd's doctoral advisor? *Robert Robinson* **3.** Which organization employs **Daniel Bernoulli** and has **Robert Robinson** as a member? *Russian Academy of Sciences* | Which organization employs **Johann Bernoulli**'s doctoral student and has **Alexander R. Todd, Baron Todd**'s doctoral advisor as a member? *Russian Academy of Sciences* |
| D | | **1.** Who is the head of state of **Kingdom of Bulgaria**? *Ferdinand I of Bulgaria* **2.** Who is the mother of **Ferdinand I of Bulgaria**? *Princess Clémentine, Princess of Koháry* **3.** Who taught **Princess Clémentine, Princess of Koháry**? *Jules Michelet* **4.** When did **Jules Michelet** begin residing in **Arathon**? *June 1852* | When did the person who taught the mother of the head of state of **Kingdom of Bulgaria** begin residing in **Arathon**? *June 1852* |
| E | | **1.** What country is **Franz Xaver Winterhalter** a citizen of? *German Empire* **2.** Who is a relative of **Princess Louise of Saxe-Gotha-Altenburg**? *Princess Margaret of Connaught* **3.** Who is the head of state of the **German Empire** whose godparent is **Princess Margaret of Connaught**? *William I, German Emperor* **4.** Which conflict did **William I, German Emperor** participate in? *Napoleonic Wars* | Which conflict did the head of state of the country **Franz Xaver Winterhalter** is a citizen of, whose godparent is a relative of **Princess Louise of Saxe-Gotha-Altenburg**, participate in? *Napoleonic Wars* |
| F | | **1.** Who won **Matteucci Medal**? *Philipp Lenard* **2.** Who was **Philipp Lenard**'s doctoral advisor? *Robert Bunsen* **3.** Who is **Henry Edward Armstrong**'s employer? *University of London* **4.** Who is both a student of **Robert Bunsen** and a director or manager at **University of London**? *Henry Enfield Roscoe* | Who is both a student of the doctoral advisor of the winner of **Matteucci Medal** and a director or manager at **Henry Edward Armstrong**'s employer? *Henry Enfield Roscoe* |

Table 3: **Multi-hop Question Reasoning Graphs and Example Questions from SYNTHWORLD-RM.** Motifs in our fact triplet graph represent recurring subgraph patterns of triplet facts that form single-hop questions, which can be composed into multi-hop questions. SYNTHWORLDS follows the same multi-hop reasoning structures as the MuSiQue dataset Trivedi et al. (2022).

## B.8 QUALITATIVE CORPORA EXAMPLES

### Robert Silverberg (Q314553)

Robert Silverberg (born 15 January 1935) is an author, novelist, science fiction writer, screenwriter and writer whose work is primarily in the science fiction genre. His given name is Robert and he began his professional career in 1955.

Silverberg was born in Brooklyn and continues to reside there. He speaks English, which is his native language and the language in which he writes. His religion is Judaism. He has cited Jack Vance and Roger Zelazny as influences on his work.

Over the course of his career Silverberg has received several awards. He was awarded Hugo Award for Best Novella in 1969, Locus Award for Best Fantasy Novel in 1981 and Locus Award for Best Novella in 1988; he received Science Fiction and Fantasy Hall of Fame on 1 January 1999 and Damon Knight Memorial Grand Master Award in 2004.

He has also been nominated for numerous literary honors, including Hugo Award for Best Novel and Hugo Award for Best Short Story in 1970, Locus Award for Best Short Story in 1972, the Locus Award for Best Novel in 1973, Hugo Award for Best Novella in 1975, Locus Award for Best Fantasy Novel in 1985, Locus Award for Best Science Fiction Novel in 1987, Locus Award for Best Novella in 1999 and Locus Award for Best Novelette in 1990.

Silverberg is described by Obálky knih.

### Yardley Raleth Quor

Yardley Raleth Quor (born 15 January 1974) is an author, novelist, science fiction writer, screenwriter and writer whose work is primarily in the science fiction genre. His given name is Yardley and he began his professional career in 1994.

Quor was born in Myrthwood and continues to reside there. He speaks Velthar, which is his native language and the language in which he writes. His religion is Veltharion. He has cited Caelian Casado and Fythar Rees as influences on his work.

Over the course of his career Quor has received several awards. He was awarded The Storyteller's Legacy in 2008, The Literary Lantern in 2020 and The Storyteller's Connection in 2027; he received Exceptional Merit Recognition on 1 January 2038 and The Page Pen Award in 2043.

He has also been nominated for numerous literary honors, including The Prose Pursuit and The Wordsmith's Triumph in 2009, Echoes of Words in 2011, the Paper Pathway Award in 2012, The Storyteller's Legacy in 2014, The Literary Lantern in 2024, The Narrative Jewel in 2026, The Storyteller's Connection in 2038 and The Inked Imagination in 2029.

Quor is described by DataGalaxy.

### Mumbai (Q1156)

Mumbai is a large urban centre on the continent Asia. It functions as the state capital and is classified as a city, a metropolis and a megacity; it is also recognized as a locality and as a business cluster, reflecting a geographic concentration of interconnected businesses in a particular field.

The settlement began in 1507. Over its history Mumbai has been within different sovereign states: it lay in Kingdom of England from 11 May 1661 until 27 March 1668, and later lay in British Raj from 28 June 1858 until 14 August 1947.

Mumbai maintains formal twinning arrangements with several other administrative bodies. It is twinned with London, Yokohama, Jakarta and Busan, and with Honolulu—the partnership with Honolulu began on 20 January 1970.

Since 2019, Mumbai has been a member of the network Creative Cities Network.

The city appears in a range of published sources. It is described in the Brockhaus and Efron Encyclopedic Dictionary (a version, edition or translation), in the Sytin Military Encyclopedia (an encyclopedic dictionary), in Jewish Encyclopedia of Brockhaus and Efron (present in ethnoreligious group, nation and people encyclopedias), and in The Nuttall Encyclopædia (a literary work).

### Crescendo

Crescendo is a large urban centre on the continent Nystoria. It functions as the state capital and is classified as a city, a metropolis and a megacity; it is also recognized as a locality and as a business cluster, reflecting a geographic concentration of interconnected businesses in a particular field.

The settlement began in 1546. Over its history Crescendo has been within different sovereign states: it lay in Kytarathia from 11 May 1700 until 27 March 1707, and later lay in Lumeria from 28 June 1897 until 14 August 1986.

Crescendo maintains formal twinning arrangements with several other administrative bodies. It is twinned with Calidore, Celestport, Eldoria and Horizon Bay, and with Jaspis—the partnership with Jaspis began on 20 January 2009.

Since 2058, Crescendo has been a member of the network SyncSphere.

The city appears in a range of published sources. It is described in the Dreamt Compilation (a version, edition or translation), in the Factoid Fount (an encyclopedic dictionary), in Qylarans (present in ethnoreligious group, nation and people encyclopedias), and in The Midnight Library (a literary work).

## B.9 PROMPTS FOR CORPORA CONSTRUCTION

---

**Generate Synthetic Names**

Give me `{{ num_names }}` fictional names for an entity X that is an instance of the following wikidata entity(ies):

> EXAMPLE INPUT INSTANCE OF INFORMATION
> - business cluster (geographic concentration of interconnected businesses in a particular field)
> - city (large human settlement)

Your response should be a list of comma separated values, eg: 'foo, bar, baz' or 'foo,bar,baz' DO NOT include any other text in your response. DO NOT reference anything that already exists in the real world.

---

**Generate Synthetic Name with Substring Relation**

Given that the following facts related to the entity X are true:

> EXAMPLE INPUT RELATED FACTS
> Cycle Ridge is a:
> - big city (city with a population of at least 100,000)
> - city (large human settlement)
> - cycling city (city designed for bicycle traffic)
>
> Fact: Cycle Ridge → location → entity X

Give me a fictional name for the entity X that is an example of a:

> EXAMPLE INPUT INSTANCE OF INFORMATION
> - public research university (type of higher learning institution; research university predominantly funded by public means)

Entity X's name is likely to consist of the names of entities that it is connected to.
Your response should be a single name for entity X on one line. DO NOT include any other text in your response. DO NOT reference anything that already exists in the real world.

---

**Generate Synth-Mapped and Real-Mapped Pages**

**System prompt:**
You are a clear, neutral, and professional writer at the level expected for Wikipedia articles: precise, informative, and fluent, without unnecessary complexity.

---

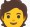 **User:**
Given a page title, the page entity information (Wikidata instance of), and a set of facts between the page title and other entities, write a high-quality Wikipedia-style article.

You will be given the following information:
- Page title
- Definitions for relation labels in page facts
- Definitions for instance-of information about related entities in page facts
- Page facts
- Instance-of information about related entities in page facts

Your task is to produce an article that uses facts faithfully, organizes them into clear prose, and avoids contradictions.
REQUIREMENTS:
- Mention every fact faithfully; do not add or invent information.
- Only use proper nouns that appear as entity names in the Page facts section.
- Organize into thematic paragraphs.

RULES FOR INTERPRETING FACTS:
- Facts are always written as **Subject → Relation → Object**.
- Interpret the relation **relative to the Subject**.
- Examples:
- *Albert Einstein → student → Nathan Rosen* → Nathan Rosen was a student of Albert Einstein.
- *Albert Einstein → student of → Alfred Kleiner* → Albert Einstein was a student of Alfred Kleiner.

FACT RELATIONS AND DIRECTIONS:
- ALWAYS follow the exact meanings of the relation labels in "Definitions for relation labels in page facts."
- NEVER invert the direction for asymmetric relations (e.g., student/student of, parent/child, advisor/doctoral student, employer/employee).
- It can be easy to get this wrong—so check the label carefully and preserve its direction exactly.
- Do not generate contradictory statements.
- Normalize symmetric relations (e.g., sibling, spouse, collaborator) into one set, de-duplicate, and group entities naturally in one or more sentences for readability.

READABILITY/WRITING STYLE:
- Do not introduce speculative context, dates, regions, or concepts.
- Do not repeat the same facts in the writing.
- NEVER write "instance of" in the writing.
- If gender is not provided, always use they/them/their.
- If gender is provided, reflect it through pronouns (he/she/they) and NOT an explit fact (to keep the writing natural and fluent).
- Vary sentence structure; avoid presenting every fact as an isolated clause or sentence.
- Group related facts into paragraphs rather than listing them line by line.
- Use connective phrasing for smoother flow (e.g., "Alongside his architectural work, he also painted. . . ").
- When presenting multiple things, use natural connectors such as "among them," "including," or "as well as" instead of flat lists.
- Break long enumerations across sentences for readability.
- Make grammatical adjustments (articles, capitalization, punctuation) for natural flow.
- Use light connective narration ("They were part of a large family. . . ") for readability.

OUTPUT: Return only the plain text article string (no Markdown).

Begin!

Page Title:   Yardley Raleth Quor

Yardley Raleth Quor   is an instance of the following entities:

EXAMPLE INPUT ON YARDLEY RALETH QUOR
- Person

### Definitions for relation labels in page facts
IMPORTANT: Use the defintions below to correctly understand the page facts.

EXAMPLE INPUT ON YARDLEY RALETH QUOR
- "award received": award or recognition received by a person, organization or creative work
- "date of birth": date on which the subject was born
- "described by source": work where this item is described
- "genre": creative work's genre or an artist's field of work (P101). Use main subject (P921) to relate creative works to their topic
- "given name": first name or another given name of this person; values used with the property should not link disambiguations nor family names
- "influenced by": this person, idea, etc. is informed by that other person, idea, etc., e.g. "Heidegger was influenced by Aristotle"
- "languages spoken, written or signed": language(s) that a person or a people speaks, writes or signs, including the native language(s)
- "native language": language or languages a person has learned from early childhood
- "nominated for": award nomination received by a person, organisation or creative work (inspired from "award received" (Property:P166))
- "occupation": occupation of a person; see also "field of work" (Property:P101), "position held" (Property:P39)
- "place of birth": most specific known birth location of a person, animal or fictional character
- "religion or worldview": religion of a person, organization or religious building, or associated with this subject
- "residence": the place where the person is or has been, resident
- "sex or gender": sex or gender identity of human or animal. For human: male, female, non-binary, intersex, transgender female, transgender male, agender, etc. For animal: male organism, female organism. Groups of same gender use subclass of (P279)
- "work period (start)": start of period during which a person or group flourished (fl. = "floruit") in their professional activity
- "writing language": language in which the writer has written their work

### Definitions for instance-of information

EXAMPLE INPUT ON YARDLEY RALETH QUOR
- "award": something given to a person or a group of people to recognize their merit or excellence
- "ethnic religion": religion defined by the ethnicity of its adherents
- "language": particular system of communication, often named for the region or peoples that use it
- "lifestyle": interests, opinions, behaviours, and behavioural orientations of an individual, group, or culture
- "literary award": award for authors and literary associations
- "male given name": given name usually meant for boys and men
- "modern language": language in current use
- "natural language": language naturally spoken by humans, as opposed to "constructed" and "formal" languages
- "religion": social-cultural system
- "web portal": website that integrates applications, processes and services

### Page facts (subject → relation property → object)
IMPORTANT: Facts are always written in the form Subject → Relation → Object. The relation definition is expressed relative to the Subject (the entity on the left). Always resolve the meaning by starting from the subject.

EXAMPLE INPUT ON YARDLEY RALETH QUOR
Yardley Raleth Quor → award received → Exceptional Merit Recognition
- point in time → 2038-01-01
Yardley Raleth Quor → award received → The Literary Lantern
- point in time → 2020 (year)
Yardley Raleth Quor → award received → The Page Pen Award
- point in time → 2043 (year)
Yardley Raleth Quor → award received → The Storyteller's Connection
- point in time → 2027 (year)
Yardley Raleth Quor → award received → The Storyteller's Legacy
- point in time → 2008 (year)
Yardley Raleth Quor → date of birth → 1974-01-15
Yardley Raleth Quor → described by source → DataGalaxy
Yardley Raleth Quor → genre → science fiction
Yardley Raleth Quor → given name → Yardley
Yardley Raleth Quor → influenced by → Caelian Casado
Yardley Raleth Quor → influenced by → Fythar Rees
Yardley Raleth Quor → languages spoken, written or signed → Velthar
Yardley Raleth Quor → native language → Velthar
Yardley Raleth Quor → nominated for → Echoes of Words
- point in time → 2011 (year)
Yardley Raleth Quor → nominated for → Paper Pathway Award
- point in time → 2012 (year)
Yardley Raleth Quor → nominated for → The Inked Imagination
- point in time → 2029 (year)
Yardley Raleth Quor → nominated for → The Literary Lantern
- point in time → 2024 (year)
Yardley Raleth Quor → nominated for → The Narrative Jewel
- point in time → 2026 (year)
Yardley Raleth Quor → nominated for → The Prose Pursuit
- point in time → 2009 (year)
Yardley Raleth Quor → nominated for → The Storyteller's Connection
- point in time → 2038 (year)
Yardley Raleth Quor → nominated for → The Storyteller's Legacy
- point in time → 2014 (year)
Yardley Raleth Quor → nominated for → The Wordsmith's Triumph
- point in time → 2009 (year)
Yardley Raleth Quor → occupation → author
Yardley Raleth Quor → occupation → novelist
Yardley Raleth Quor → occupation → science fiction writer
Yardley Raleth Quor → occupation → screenwriter
Yardley Raleth Quor → occupation → writer
Yardley Raleth Quor → place of birth → Myrthwood
Yardley Raleth Quor → religion or worldview → Veltharion
Yardley Raleth Quor → residence → Myrthwood
Yardley Raleth Quor → sex or gender → male
Yardley Raleth Quor → work period (start) → 1994 (year)
Yardley Raleth Quor → writing language → Velthar

### Instance-of information about related entities in page facts (entity: wikidata entities which the entity is an instance of)

EXAMPLE INPUT ON YARDLEY RALETH QUOR
- "Caelian Casado": person
- "DataGalaxy": multimedia database and web portal
- "Echoes of Words": literary award
- "Exceptional Merit Recognition": award
- "Fythar Rees": person
- "Myrthwood": borough of global city, borough of port settlement, borough of city, borough of big city, borough of largest city, borough of metropolis and borough of megacity, city in the sovereign state, city in the constitutional republic, city in the federal republic, city in the superpower, city in the country and city in the historical unrecognized state and consolidated city-county
- "Paper Pathway Award": award for best book (by genre) and literary award
- "The Inked Imagination": literary award
- "The Literary Lantern": award for best book (by genre)
- "The Narrative Jewel": literary award
- "The Page Pen Award": literary award
- "The Prose Pursuit": literary award
- "The Storyteller's Connection": literary award
- "The Storyteller's Legacy": literary award
- "The Wordsmith's Triumph": literary award
- "Velthar": language, modern language and natural language
- "Veltharion": ethnic religion, lifestyle and religion
- "Yardley": male given name

## Page Content

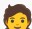 **Assistant:**

Yardley Raleth Quor (born 15 January 1974) is an author, novelist, science fiction writer, screenwriter and writer whose work is primarily in the science fiction genre. His given name is Yardley and he began his professional career in 1994.

Quor was born in Myrthwood and continues to reside there. He speaks Velthar, which is his native language and the language in which he writes. His religion is Veltharion. He has cited Caelian Casado and Fythar Rees as influences on his work.

Over the course of his career Quor has received several awards. He was awarded The Storyteller's Legacy in 2008, The Literary Lantern in 2020 and The Storyteller's Connection in 2027; he received Exceptional Merit Recognition on 1 January 2038 and The Page Pen Award in 2043.

He has also been nominated for numerous literary honors, including The Prose Pursuit and The Wordsmith's Triumph in 2009, Echoes of Words in 2011, the Paper Pathway Award in 2012, The Storyteller's Legacy in 2014, The Literary Lantern in 2024, The Narrative Jewel in 2026, The Storyteller's Connection in 2038 and The Inked Imagination in 2029.

Quor is described by DataGalaxy.

---

**User:**

Your job is to now rewrite the answer you provided above, but instead of responding directly with the text, transform the text by replacing related entity mentions with linked references using markdown and Jinja-style expressions: '[Entity Text]({{Entity_ID }})'

You will be given the following:
1. Entity Reference JSON: A mapping of entity IDs to their name labels for each related entity.

For example, given:

```json
{
  "Q1397": "Ohio (U.S. state)",
  "Q30": "United States (sovereign state)"
}
```

You would transform:
"She was born in Ohio, USA."
To:
"She was born in [Ohio]({{ Q1397 }}), [USA]({{ Q30 }})."

IMPORTANT:
- Preserve all grammar, punctuation, and readability from the original text.
- The text and the spacing outside of the links should stay the same same.
- The text without the links should still be fluent and have proper grammar and punctuation.
- Only link proper nouns (capitalized entities like names, places, organizations)
- Only create links for entities that exist in the Entity Reference JSON. Never invent IDs or assume availability.
- As a general rule of thumb, link only the first occurrence of an entity in the text of the article.
- Links should not contain leading or trailing spaces within the square brackets, e.g., use '[North America]({{ Q49 }})', not '[North America ]({{ Q49 }})'.
- DO NOT add square brackets to terms that are not in the Entity Reference JSON, e.g., "Mercedes-Benz is a [car manufacturer] founded in 1926" is incorrect.

BAD EXAMPLES:
- "They travelled to [South America]({{ Q30 }})" when Q30 refers to "United States"
- "They visited [Paris]({{ Q90 }})" when Q90 is not in the provided JSON.

Begin!

## Entity Reference JSON:

EXAMPLE INPUT ON YARDLEY RALETH QUOR

```
{
  "Q1860": "Velthar",
  "Q9268": "Veltharion",
  "Q18419": "Myrthwood",
  "Q255032": "The Prose Pursuit",
  "Q295406": "Fythar Rees",
  "Q312632": "Caelian Casado",
  "Q549884": "The Storyteller's Legacy",
  "Q595998": "Paper Pathway Award",
  "Q607354": "The Literary Lantern",
  "Q901462": "The Page Pen Award",
  "Q1056251": "The Wordsmith's Triumph",
  "Q2576795": "The Narrative Jewel",
  "Q4927937": "Yardley",
  "Q17008964": "The Inked Imagination",
  "Q17008972": "The Storyteller's Connection",
  "Q17008985": "Echoes of Words",
  "Q48961500": "Exceptional Merit Recognition",
  "Q67311526": "DataGalaxy"
}
```

## New Output:

🖥 **Assistant:**

EXAMPLE OUTPUT GPT-5-MINI

Yardley Raleth Quor (born 15 January 1974) is an author, novelist, science fiction writer, screenwriter and writer whose work is primarily in the science fiction genre. His given name is [Yardley]({{ Q4927937 }}) and he began his professional career in 1994.

Quor was born in [Myrthwood]({{ Q18419 }}) and continues to reside there. He speaks [Velthar]({{ Q1860 }}), which is his native language and the language in which he writes. His religion is [Veltharion]({{ Q9268 }}). He has cited [Caelian Casado]({{ Q312632 }}) and [Fythar Rees]({{ Q295406 }}) as influences on his work.

Over the course of his career Quor has received several awards. He was awarded [The Storyteller's Legacy]({{ Q549884 }}) in 2008, [The Literary Lantern]({{ Q607354 }}) in 2020 and [The Storyteller's Connection]({{ Q17008972 }}) in 2027; he received [Exceptional Merit Recognition]({{ Q48961500 }}) on 1 January 2038 and [The Page Pen Award]({{ Q901462 }}) in 2043.

He has also been nominated for numerous literary honors, including [The Prose Pursuit]({{ Q255032 }}) and [The Wordsmith's Triumph]({{ Q1056251 }}) in 2009, [Echoes of Words]({{ Q17008985 }}) in 2011, the [Paper Pathway Award]({{ Q595998 }}) in 2012, The Storyteller's Legacy in 2014, The Literary Lantern in 2024, [The Narrative Jewel]({{ Q2576795 }}) in 2026, The Storyteller's Connection in 2038 and [The Inked Imagination]({{ Q17008964 }}) in 2029.

Quor is described by [DataGalaxy]({{ Q67311526 }}).

🧑 **User:**

The previous response was perturbed, where names of entities and timestamp values were intentionally changed.

The actual page is about  Robert Silverberg  but in the above text, it was renamed to

 Yardley Raleth Quor .

Your task is to now restore the above text by replacing all modified names and dates with their original values, using the provided JSON mapping.

The rest of the text must remain exactly as it is — only change the names and dates that appear in the mapping.

For example given:

```json
{
  "John Smith": "Elon Musk",
  "2023-07-14": "2020-07-14",
  "Acme Corp": "SpaceX"
```

```
    }
    ```
```

and text that said:

"On July 27, 2023, John Smith announced that Acme Corp would begin developing a reusable rocket. Smith was a leader in the company."

You would rewrite this to:

"On July 27, 2020, Elon Musk announced thatSpaceX would begin developing a reusable rocket. Musk was a leader in the company."

IMPORTANT.

The final output must not contain any perturbed entity names and perturbed timestamp values. Outside of the renaming, the rest of the text should stay the exact same with the links are still included.

Do not rephrase or modify the surrounding text in any way.

Begin!

EXAMPLE INPUT ON YARDLEY RALETH QUOR

```
{
  "Q1860": "Velthar",
  "Q9268": "Veltharion",
  "Q18419": "Myrthwood",
  "Q255032": "The Prose Pursuit",
  "Q295406": "Fythar Rees",
  "Q312632": "Caelian Casado",
  "Q549884": "The Storyteller's Legacy",
  "Q595998": "Paper Pathway Award",
  "Q607354": "The Literary Lantern",
  "Q901462": "The Page Pen Award",
  "Q1056251": "The Wordsmith's Triumph",
  "Q2576795": "The Narrative Jewel",
  "Q4927937": "Yardley",
  "Q17008964": "The Inked Imagination",
  "Q17008972": "The Storyteller's Connection",
  "Q17008985": "Echoes of Words",
  "Q48961500": "Exceptional Merit Recognition",
  "Q67311526": "DataGalaxy"
}
```

## Rewritten Output:

**🖥 Assistant:**

EXAMPLE OUTPUT GPT-5-MINI

Robert Silverberg (born 15 January 1935) is an author, novelist, science fiction writer, screenwriter and writer whose work is primarily in the science fiction genre. His given name is [Robert]({{ Q4927937 }}) and he began his professional career in 1955.

Silverberg was born in [Brooklyn]({{ Q18419 }}) and continues to reside there. He speaks [English]({{ Q1860 }}), which is his native language and the language in which he writes. His religion is [Judaism]({{ Q9268 }}). He has cited [Jack Vance]({{ Q312632 }}) and [Roger Zelazny]({{ Q295406 }}) as influences on his work.

Over the course of his career Silverberg has received several awards. He was awarded [Hugo Award for Best Novella]({{ Q549884 }}) in 1969, [Locus Award for Best Fantasy Novel]({{ Q607354 }}) in 1981 and [Locus Award for Best Novella]({{ Q17008972 }}) in 1988; he received [Science Fiction and Fantasy Hall of Fame]({{ Q48961500 }}) on 1 January 1999 and [Damon Knight Memorial Grand Master Award]({{ Q901462 }}) in 2004.

He has also been nominated for numerous literary honors, including [Hugo Award for Best Novel]({{ Q255032 }}) and [Hugo Award for Best Short Story]({{ Q1056251 }}) in 1970, [Locus Award for Best Short Story]({{ Q17008985 }}) in 1972, the [Locus Award for Best Novel]({{ Q595998 }}) in 1973, Hugo Award for Best Novella in 1975, Locus Award for Best Fantasy Novel in 1985, [Locus Award for Best Science Fiction Novel]({{ Q2576795 }}) in 1987, Locus Award for Best Novella in 1999 and [Locus Award for Best Novelette]({{ Q17008964 }}) in 1990.

Silverberg is described by [Obálky knih]({{ Q67311526 }}).

---

**Validate Page Novelty**

**System prompt:**
You are a master detective that identifies the main subject of a given page. Entity names in pages have been replaced with fictional ones to create a parallel fictional world. Your job is to identify the original real-world entity by analyzing patterns in dates, events, relationships, achievements, and biographical details.

---

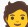 **User:**
The following page is about a real-world wikidata entity, but some or all names have been changed to fictional ones. Identify the original main subject.
Hint: The entity currently named in the page is fictional - you need to determine which real person/organization/thing this fictional entity represents.
Your response starts with "Thought: ", where you will methodically break down the reasoning process, illustrating how you arrive at conclusions. Then, based on your reasoning, conclude with the prefix "Answer: " to present a concise, definitive response, devoid of additional elaborations (e.g., "Answer: Barack Obama").

Page Content:

INPUT SYNTH-MAPPED PAGE

Thought:

---

### B.10 PROMPTS FOR MULTI-HOP QA CONSTRUCTION FROM FACTS

**Get Page Facts**

**System prompt:**
You are an advanced reading comprehension assistant. Your task is to analyze a text passage and extract specific information to fill in triplet templates with placeholders marked as <ANSWER>.

---

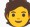 **User:**
Given a page and a JSON mapping of partial facts (indicated by <ANS> placeholders), use the page content to extract the missing information.
Return your output as a list of answers for all triplets. For nested triplets (containing multiple <ANS> placeholders), ensure all parts are supported by the text.

Guidelines:
- Return empty array

if no relevant information is found
- For nested triplets, only include complete matches where all placeholder values are found
- If multiple valid answers exist, include all of them
- Use the relation descriptions to help you understand the meaning of the triplet relations. Triplets are always in the form of subject → relation → object.

Example:
Page content:
Pavel Cherenkov held roles as a nuclear physicist and a general physicist, and over the course of his career he received several honors, including the Nobel Prize in Physics, the Order of Lenin, the Order of the Red Banner of Labour and the Hero of Socialist Labour. He was nominated for the Nobel Prize in Physics in 1955 and later received that prize in 1958.

Relation descriptions:
- "award received": award or recognition received by a person, organization or creative work
- "nominated for": award nomination received by a person, organisation or creative work (inspired from "award received" (Property:P166))
- "occupation": occupation of a person; see also "field of work" (Property:P101), "position held" (Property:P39)

---

Partial fact templates:

```json
{
"T1": Pavel Cherenkov -> occupation -> <ANS>
"T2": Pavel Cherenkov -> award received -> <ANS>
"T3": Pavel Cherenkov -> award received -> <ANS1> AND <ANS1> -> point in time -> <
    ANS2>
"T4": Pavel Cherenkov -> nominated for -> <ANS1> AND <ANS1> -> point in time -> <
    ANS2>
}
```

Output:

```json
{
"T1": ["nuclear physicist", "physicist"]
"T2": ["Nobel Prize in Physics", "Order of Lenin", "Order of the Red Banner of
    Labour", "Hero of Socialist Labour"]
"T3": [["Nobel Prize in Physics", "1955"]]
"T4": [["Nobel Prize in Physics", "1958"]]
}
```

Output format:
Return only the JSON object with extracted answers. Use empty arrays for triplets where no information is found in the text.

Begin!
Page content:

EXAMPLE INPUT ON YARDLEY RALETH QUOR

Yardley Raleth Quor (born 15 January 1974) is an author, novelist, science fiction writer, screenwriter and writer whose work is primarily in the science fiction genre. His given name is Yardley and he began his professional career in 1994.

Quor was born in Myrthwood and continues to reside there. He speaks Velthar, which is his native language and the language in which he writes. His religion is Veltharion. He has cited Caelian Casado and Fythar Rees as influences on his work.

Over the course of his career Quor has received several awards. He was awarded The Storyteller's Legacy in 2008, The Literary Lantern in 2020 and The Storyteller's Connection in 2027; he received Exceptional Merit Recognition on 1 January 2038 and The Page Pen Award in 2043.

He has also been nominated for numerous literary honors, including The Prose Pursuit and The Wordsmith's Triumph in 2009, Echoes of Words in 2011, the Paper Pathway Award in 2012, The Storyteller's Legacy in 2014, The Literary Lantern in 2024, The Narrative Jewel in 2026, The Storyteller's Connection in 2038 and The Inked Imagination in 2029.

Quor is described by DataGalaxy.

Relation descriptions:

EXAMPLE INPUT ON YARDLEY RALETH QUOR

- award received: award or recognition received by a person, organization or creative work
- date of birth: date on which the subject was born
- described by source: work where this item is described
- genre: creative work's genre or an artist's field of work (P101). Use main subject (P921) to relate creative works to their topic
- given name: first name or another given name of this person; values used with the property should not link disambiguations nor family names
- influenced by: this person, idea, etc. is informed by that other person, idea, etc., e.g. "Heidegger was influenced by Aristotle"
- languages spoken, written or signed: language(s) that a person or a people speaks, writes or signs, including the native language(s)
- native language: language or languages a person has learned from early childhood
- nominated for: award nomination received by a person, organisation or creative work (inspired from "award received" (Property:P166))
- occupation: occupation of a person; see also "field of work" (Property:P101), "position held" (Property:P39)
- place of birth: most specific known birth location of a person, animal or fictional character
- religion or worldview: religion of a person, organization or religious building, or associated with this subject
- residence: the place where the person is or has been, resident
- sex or gender: sex or gender identity of human or animal. For human: male, female, non-binary, intersex, transgender female, transgender male, agender, etc. For animal: male organism, female organism. Groups of same gender use subclass of (P279)
- work period (start): start of period during which a person or group flourished (fl. = "floruit") in their professional activity
- writing language: language in which the writer has written their work

Partial fact templates:

```
EXAMPLE INPUT ON YARDLEY RALETH QUOR

{
  "T0": "Yardley Raleth Quor -> award received -> <ANS1> AND <ANS1> -> point in
      time -> <ANS2>",
  "T1": "Yardley Raleth Quor -> date of birth -> <ANS>",
  "T2": "Yardley Raleth Quor -> described by source -> <ANS>",
  "T3": "Yardley Raleth Quor -> genre -> <ANS>",
  "T4": "Yardley Raleth Quor -> given name -> <ANS>",
  "T5": "Yardley Raleth Quor -> influenced by -> <ANS>",
  "T6": "Yardley Raleth Quor -> languages spoken, written or signed -> <ANS>",
  "T7": "Yardley Raleth Quor -> native language -> <ANS>",
  "T8": "Yardley Raleth Quor -> nominated for -> <ANS1> AND <ANS1> -> point in
      time -> <ANS2>",
  "T9": "Yardley Raleth Quor -> occupation -> <ANS>",
  "T10": "Yardley Raleth Quor -> place of birth -> <ANS>",
  "T11": "Yardley Raleth Quor -> religion or worldview -> <ANS>",
  "T12": "Yardley Raleth Quor -> residence -> <ANS>",
  "T13": "Yardley Raleth Quor -> sex or gender -> <ANS>",
  "T14": "Yardley Raleth Quor -> work period (start) -> <ANS>",
  "T15": "Yardley Raleth Quor -> writing language -> <ANS>"
}
```

Output:

💻 **Assistant:**

```
EXAMPLE OUTPUT ON GPT-5-MINI

{
  "T0": [
    ["The Storyteller's Legacy", "2008"],
    ["The Literary Lantern", "2020"],
    ["The Storyteller's Connection", "2027"],
    ["Exceptional Merit Recognition", "1 January 2038"],
    ["The Page Pen Award", "2043"]
  ],
  "T1": ["15 January 1974"],
  "T2": ["DataGalaxy"],
  "T3": ["science fiction"],
  "T4": ["Yardley"],
  "T5": ["Caelian Casado", "Fythar Rees"],
  "T6": ["Velthar"],
  "T7": ["Velthar"],
  "T8": [
    ["The Prose Pursuit", "2009"],
    ["The Wordsmith's Triumph", "2009"],
    ["Echoes of Words", "2011"],
    ["Paper Pathway Award", "2012"],
    ["The Storyteller's Legacy", "2014"],
    ["The Literary Lantern", "2024"],
    ["The Narrative Jewel", "2026"],
    ["The Storyteller's Connection", "2038"],
    ["The Inked Imagination", "2029"]
  ],
  "T9": ["author", "novelist", "science fiction writer", "screenwriter", "writer
      "],
  "T10": ["Myrthwood"],
  "T11": ["Veltharion"],
  "T12": ["Myrthwood"],
  "T13": [],
  "T14": ["1994"],
  "T15": ["Velthar"]
}
```

## Generate Single Hop Questions from Facts

**System prompt:**

You are an expert annotator for a question answering dataset. Your task is to generate high-quality, fact-based questions from the provided facts and relation descriptions.

---

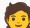 **User:**
You will be given a specific graph structure defining how questions are generated from facts (i.e., entities/values (E) and relations (R)).
Each E can represent either an entity (person, place, thing) or a value (date, number, text).

### Structure:

> STRUCTURE FOR GRAPH A (WOULD CHANGE FOR OTHER GRAPHS)
> Q1: E1 → R1 → E2 <ANS1>
> Q2: E2 → R2 → E3 <ANS2>
> Where E1, E2, E3 are entities/values and R1, R2 are relations.
> <ANS1> and <ANS2> are different answers to Q1 and Q2 respectively.

### Relation Types:
- Simple Relations: A direct relationship between two entities/values (entity → relation → entity)
- Qualified Relations: When a relation needs additional context (time, location, role, etc.):
- 'Entity → [BaseRelation → Qualifier → Attribute] → Value'
- Interpretation: The Attribute of Qualifier's BaseRelation to Entity is Value
- Example:
- 'Paris → [mayor → Anne Hidalgo → start time] → 2014-04-05'
- Means: "The start time of Anne Hidalgo's role as mayor of Paris is 2014-04-05"
- Question: "When did Anne Hidalgo become mayor of Paris?"
- Facts are written in the form Subject → Relation → Object. The relation definition is expressed relative to the Subject (the entity on the left). Always resolve the meaning by starting from the subject.

### Requirements:
- Each question must be natural, fluent English and have a single, unambiguous correct answer.
- The answer to each question is exactly the entity/value tagged with <ANS>.
- The subject entity (the entity before → R...) must appear explicitly in the question text to ensure clarity.
- Phrase time-based relations naturally ("When did...?", "On what date...?", "In what year...?") matching the granularity of the <ANS> (date/year/etc.).
- Do not copy awkward relation phrasing verbatim if a more natural form exists ("Where was X born?", not "What is the place of birth of X?").
- Do not include <ANS> verbatim in the question text — the question must point to <ANS> naturally without revealing it.

#### Relations
- Use the relation wording from the Question Facts as the basis for your question, rephrasing only if needed for natural English.
- If the entity type is unclear (e.g., the relation description lists multiple possible types such as country or region), avoid inventing context (e.g., ask "What shares a border with X?" instead of "What country borders X?").
- Avoid using the word "entity" in the question text — questions should always sound natural.

### Output Format:
Respond only with questions in this JSON format:

```json
{
  "Q1": "Question 1",
  "Q2": "Question 2",
  "QN": "Question N"
}
```

Do not include explanations, comments, or text outside the JSON object.

### Example:

DEMONSTRATION FOR GRAPH A (WOULD CHANGE FOR OTHER GRAPHS)

Question Facts:

Q1: Stephen Hawking → place of birth → United Kingdom <ANS1>

Q2: United Kingdom → capital → London <ANS2>

Relation Descriptions:

- place of birth: most specific known birth location of a person, animal or fictional character
- capital: seat of government of a country, province, state or other type of administrative territorial entity

Output:

```json
{{
"Q1": "Where was Stephen Hawking born?",
"Q2": "What is the capital of the United Kingdom?"
}}
```

Begin!
Question Facts:

EXAMPLE INPUT FOR GRAPH A

Question Facts:

Q1: Jorith Luque → educated at → The Artistic Exchange <ANS1>

Q2: The Artistic Exchange → [founded by → Merith Watts → point in time] → 1864 (year) <ANS2>

Relation Descriptions:

EXAMPLE INPUT FOR GRAPH A

- "educated at": educational institution attended by subject
- "founded by": founder or co-founder of this organization, religion, place or entity
- "point in time": date something took place, existed or a statement was true; for providing time use the "refine date" property (P4241)

Output:

---

💻 **Assistant:**

EXAMPLE OUTPUT GPT-5-MINI

```json
{
  "Q1": "Where was Jorith Luque educated?",
  "Q2": "In what year did Merith Watts found The Artistic Exchange?"
}
```

## Generate Multi-hop Questions from Single-hop Questions

**System prompt:**
You are an expert annotator for a question answering dataset. Your task is to compose a coherent question from a list of decomposed questions.
Each decomposed question represents one atomic fact or relationship.
For example, given the following decomposed questions:
- Q1: Which university was Facebook launched in? → Harvard
- Q2: What city is <bridge> Harvard</bridge> located in? → Cambridge
They can be composed together into:
"Which city was Facebook launched in?"
Bridge entities are marked with <bridge> tags, each of which should be the answer of a decomposed question.
Characteristics of Good Questions:

- Fact-seeking: Questions that can be answered with a specific entity or concise explanation
- Unambiguous: Has a single, clear correct answer
- Requires comprehension: Demonstrates understanding beyond surface-level pattern matching
- Natural language: Uses conversational phrasing that sounds like something a person would ask
Characteristics of Bad Questions:
- Poorly formulated: Unclear or grammatically incorrect questions
- False presuppositions: Questions based on incorrect assumptions
- Opinion-based: Questions seeking subjective judgments rather than factual information
- Not fact-seeking: Questions that don't clearly request factual information

---

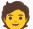 **User:**
Your will be given a list of decomposed questions with marked bridge entities. Your task is to compose a coherent question from them.

IMPORTANT The composed question SHOULD NOT include any bridge entities (i.e., those wrapped in <bridge> tags). If composed correctly, the bridge entities should not occur in the composed question.

Requirements for composing questions:
1. Use all decomposed questions: Incorporate information from all decomposed questions.
2. Preserve meaning and answer: Retain the meaning and ensure the composed question's answer is the same as the last decomposed question's answer. Do not change the answer. Rephrasing is encouraged for fluency and clarity. (Incorrect example: "Which country was Facebook launched in?" — this changes both meaning and answer.)
3. Keep it concise: Compress as much as possible without losing meaning. Prefer: "Which city was Facebook launched in?" over "Which city has a university, which Facebook was launched in?"
4. Two-sentence fallback: If the composed question becomes too long to be coherent, you may split it into two sentences (1 assertion + 1 question), connected by coreference. Use only as a last resort.
5. Answer alignment: The composed question must always have the same answer as the last decomposed question's answer.
6. Do not remove necessary details that would make the question ambiguous. This means that non-bridge entities should be included
7. Phrase time-based relations naturally ("When did...?", "On what date...?", "In what year...?") matching the granularity of the answer (date/year/etc.).

FAQ:
1. Should I paraphrase the question for clarity?
  You're encouraged to paraphrase the question to make it simple and coherent as long as the rephrased question leads to the associated answer. You do not need to use the exact same phrasing as the decomposed question, because sometimes they are awkward. E.g. replacing "terrain feature" → "mountain range", "administrative territorial entity" → "state/city/etc", "parental progenitor" → mother/father depending on the question context are all great
2. Given the hard choice, do you prefer a shorter or more coherent question?
  Being able to parse and understand the question is more important to us than its length. So if you can't retain coherency of the question while keeping it short, write a longer but coherent composed question.
3. What if the question is composable even if entities aren't exactly the same?
  There are some rare cases in which the marked bridge entity doesn't mean exactly the same, but yet are talking about the same entity. E.g.
  - Q1: Who was in charge of the US? → George Washington
  - Q2: Who was the creator of George Washington? → Donald De Lue
  Here, Q1's "George Washington" is a person, while Q2's refers to a monument of him. You can compose: "Who is the creator of the monument of the person in charge of the US?"
  But avoid nonsensical versions like: "Who is the creator of the person in charge of the US?"

If the composition would be too awkward or confusing, respond with "No composition".

Output format:
Start your answer with "Thought: ", where you reason through your decision step-by-step.

Conclude with "Question: ", followed by the composed question (or "No composition") without any modification (i.e., no formatting, no bolding, and no markup) or further explanation.

Examples:

DEMONSTRATIONS BASED ON MUSIQUE

Decomposed questions:
- Q1: Who was the first President of Namibia? → Sam Nujoma
- Q2: Who succeeded <bridge>Sam Nujoma</bridge>? → Hifikepunye Pohamba
Thought:
- Q1 tells us that Sam Nujoma was the first President of Namibia.
- Q2 asks who succeeded <bridge>Sam Nujoma</bridge>, referring to the person identified in Q1.
- Since Sam Nujoma = first President of Namibia, we can substitute that description into Q2.
Question: Who succeeded the first President of Namibia?

Decomposed questions:
- Q1: At what location did Billy Giles die? → Belfast
- Q2: What part of the UK is <bridge>Belfast</bridge>located in? → Northern Ireland
- Q3: What is the unit of currency in <bridge>Northern Ireland</bridge>? → Pound sterling
Thought:
- Q1 says Billy Giles died in Belfast.
- Q2 tells us Belfast is in Northern Ireland.
- Q3 says Northern Ireland uses Pound sterling.
- So we can describe the place where Billy Giles died as "Northern Ireland."
Question: What currency is used where Billy Giles died?

Decomposed questions:
- Q1: What is McDonaldization named after? → McDonald's
- Q2: Which state is Horndean located in? → England
- Q3: When did the first <bridge>McDonald's</bridge>open in <bridge>England</bridge>? → 1974
Thought:
- Q1 says McDonaldization is named after McDonald's.
- Q2 says Horndean is located in England.
- Q3 asks when McDonald's first opened in England.
- Since Horndean is in England, we can use that substitution to generalize the location.
Question: When was the first establishment that McDonaldization is named after opened in the country Horndean is located?

Decomposed questions:
- Q1: Who brought Louis XVI style to the court? → Marie Antoinette
- Q2: Who is the mother of <bridge>Marie Antoinette</bridge>? → Maria Theresa
- Q3: In what city did <bridge>Maria Theresa</bridge>die? → Vienna
- Q4: When did Napoleon occupy <bridge>Vienna</bridge>? → 1805
Thought:
- Q1 tells us Marie Antoinette brought Louis XVI style to the court.
- Q2 tells us her mother is Maria Theresa.
- Q3 says Maria Theresa died in Vienna.
- Q4 tells us Napoleon occupied Vienna in 1805.
- So we can ask when Napoleon occupied the city where the mother of the woman who brought Louis XVI style to the court died.
Question: When did Napoleon occupy the city where the mother of the woman who brought Louis XVI style to the court died?

Decomposed questions:
- Q1: What continent is Aruba in? → South America
- Q2: What country is Prazeres from? → Portugal
- Q3: What was the colonial holding in <bridge>South America</bridge>governed by <bridge>Portugal</bridge>? → Brazil
- Q4: How many Germans live in <bridge>Brazil</bridge>? → 5 million
Thought:
- Q1 tells us Aruba is in South America.
- Q2 tells us Prazeres is from Portugal.
- Q3 tells us Brazil was the colonial holding in South America governed by Portugal.
- Q4 says 5 million Germans live in Brazil.
- So we can ask how many Germans live in the colonial holding in Aruba's continent that was governed by Prazeres's country.
Question: How many Germans live in the colonial holding in Aruba's continent that was governed by Prazeres's country?

Begin!
Decomposed questions:

EXAMPLE INPUT SINGLE HOP QUESTIONS FOR GRAPH A

Q1: Where was Jorith Luque educated? → The Artistic Exchange
Q2: In what year did Merith Watts found <bridge> The Artistic Exchange</bridge> ? → 1864 (year)

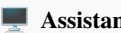 **Assistant:**

> EXAMPLE OUTPUT GPT-5-MINI
>
> Thought: Q1 says Jorith Luque was educated at The Artistic Exchange. Q2 asks when Merith Watts founded that same institution and gives the year 1864. Combine by asking when Merith Watts founded the institution where Jorith Luque was educated, avoiding the bridge name.
> Question: In what year did Merith Watts found the institution where Jorith Luque was educated?

## C EXPERIMENT DETAILS

For all experiments with Gemini-2.0-Flash with temperature=0. For all experiments with GPT-5-mini, the reasoning effort is set to default `medium`.

For the HippoRAG 2 baselines (Gutiérrez et al., 2025), we used the same model for NER and OpenIE (i.e, GPT-5-mini for GPT-5-mini experiments and Gemini-2.0-Flash and Gemini-2.0-Flash experiments). We follow (Gutiérrez et al., 2025) and use nvidia/NVEmbed-v2 (Lee et al., 2025) as the retriever. For IRCoT, we run for a maximum of 10 steps.

F1 error bars (Fig. 4) are calculated with 95% bootstrap confidence intervals. For each difficulty bucket, we resample with replacement 5,000 times, compute the mean F1 (or F1-gap) for each resample, and take the 2.5th and 97.5th percentiles as confidence bounds. Success-rate error bars (Fig. 5) are calculated using 95% Wilson score confidence intervals, computed via `statsmodels.stats.proportion.proportion_confint` with `method='wilson'` in Python.

### C.1 MULTIHOP QA PROMPTS

For the QA baselines, we follow prior work whenever possible (Trivedi et al., 2022; Gutierrez et al., 2024), including the use of prompt demonstrations.

---

**Multihop Question Answering — No Retrieval**

**System prompt:**
As an advanced question answering assistant, your task is to answer the question. Your response starts after "Thought: ", where you will methodically break down the reasoning process, illustrating how you arrive at conclusions. Conclude with "Answer: " to present a concise, definitive response, devoid of additional elaborations. Your answer should be a single entity or timestamp.

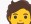 **User:**
Question: `{{ query }}`
Thought:

---

**Multihop Question Answering — Reading Comprehension (includes one demonstration)**

**System prompt:**
As an advanced question answering assistant, your task is to analyze text passages and corresponding questions meticulously. Your response starts after "Thought: ", where you will methodically break down the reasoning process, illustrating how you arrive at conclusions. Conclude with "Answer: " to present a concise, definitive response, devoid of additional elaborations. Your answer should be a single entity or timestamp.

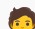 **User:**

---

The Last Horse (Spanish: El Último caballo) is a 1950 Spanish comedy film directed by Edgar Neville starring Fernando Fernán Gómez.

The University of Southampton, which was founded in 1862 and received its Royal Charter as a university in 1952, has over 22,000 students. The university is ranked in the top 100 research universities in the world in the Academic Ranking of World Universities 2010. In 2010, the THES - QS World University Rankings positioned the University of Southampton in the top 80 universities in the world. The university considers itself one of the top 5 research universities in the UK. The university has a global reputation for research into engineering sciences, oceanography, chemistry, cancer sciences, sound and vibration research, computer science and electronics, optoelectronics and textile conservation at the Textile Conservation Centre (which is due to close in October 2009.) It is also home to the National Oceanography Centre, Southampton (NOCS), the focus of Natural Environment Research Council-funded marine research.

Stanton Township is a township in Champaign County, Illinois, USA. As of the 2010 census, its population was 505 and it contained 202 housing units.

Neville A. Stanton is a British Professor of Human Factors and Ergonomics at the University of Southampton. Prof Stanton is a Chartered Engineer (C.Eng), Chartered Psychologist (C.Psychol) and Chartered Ergonomist (C.ErgHF). He has written and edited over forty books and over three hundred peer-reviewed journal papers on applications of the subject. Stanton is a Fellow of the British Psychological Society, a Fellow of The Institute of Ergonomics and Human Factors and a member of the Institution of Engineering and Technology. He has been published in academic journals including "Nature". He has also helped organisations design new human-machine interfaces, such as the Adaptive Cruise Control system for Jaguar Cars.

Finding Nemo
Theatrical release poster
Directed by Andrew Stanton
Produced by Graham Walters
Screenplay by Andrew Stanton Bob Peterson David Reynolds
Story by Andrew Stanton
Starring Albert Brooks Ellen DeGeneres Alexander Gould Willem Dafoe
Music by Thomas Newman
Cinematography Sharon Calahan Jeremy Lasky
Edited by David Ian Salter
Production company Walt Disney Pictures Pixar Animation Studios
Distributed by Buena Vista Pictures Distribution
Release date May 30, 2003 (2003-05-30)
Running time 100 minutes
Country United States
Language English
Budget $94 million
Box office $940.3 million

Question: When was Neville A. Stanton's employer founded?
Thought:

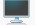 **Assistant:**
The employer of Neville A. Stanton is University of Southampton. The University of Southampton was founded in 1862.
Answer: 1862

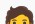 **User:**
`{{ gold_and_distractor_passages }}`
Question: `{{ query }}`
Thought:

---

**Multihop Question Answering — One-Step RAG (includes one demonstration)**

**System prompt:**
As an advanced reading comprehension assistant, your task is to answer the question given the passages. Your response starts after "Thought: ", where you will methodically break down the reasoning process, illustrating how you arrive at conclusions. Conclude with "Answer: " to present a concise, definitive response, devoid of additional elaborations, explanations or extra information. Your answer should be a single entity or timestamp.

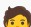 **User:**

SAME AS READING COMPREHENSION

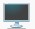 **Assistant:**
The employer of Neville A. Stanton is University of Southampton. The University of Southampton was founded in 1862.
Answer: 1862

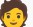 **User:**
{{ retrieved_passages }}
Question: {{ query }}
Thought:

## Multihop Question Answering — IRCoT + RAG (includes demonstration)

**System prompt:**
You serve as an intelligent assistant, adept at facilitating users through complex, multi-hop reasoning across multiple documents. This task is illustrated through demonstrations, each consisting of a document set paired with a relevant question and its multi-hop reasoning thoughts. Your task is to generate one thought for the current step, DON'T generate the whole thoughts at once! If you reach what you believe to be the final step, start with "So the answer is:" Your answer should be a single entity or timestamp.

EXAMPLE DEMONSTRATION

The Last Horse (Spanish: El Último caballo) is a 1950 Spanish comedy film directed by Edgar Neville starring Fernando Fernán Gómez.

The University of Southampton, which was founded in 1862 and received its Royal Charter as a university in 1952, has over 22,000 students. The university is ranked in the top 100 research universities in the world in the Academic Ranking of World Universities 2010. In 2010, the THES - QS World University Rankings positioned the University of Southampton in the top 80 universities in the world. The university considers itself one of the top 5 research universities in the UK. The university has a global reputation for research into engineering sciences, oceanography, chemistry, cancer sciences, sound and vibration research, computer science and electronics, optoelectronics and textile conservation at the Textile Conservation Centre (which is due to close in October 2009.) It is also home to the National Oceanography Centre, Southampton (NOCS), the focus of Natural Environment Research Council-funded marine research.

Stanton Township is a township in Champaign County, Illinois, USA. As of the 2010 census, its population was 505 and it contained 202 housing units.

Neville A. Stanton is a British Professor of Human Factors and Ergonomics at the University of Southampton. Prof Stanton is a Chartered Engineer (C.Eng), Chartered Psychologist (C.Psychol) and Chartered Ergonomist (C.ErgHF). He has written and edited over forty books and over three hundred peer-reviewed journal papers on applications of the subject. Stanton is a Fellow of the British Psychological Society, a Fellow of The Institute of Ergonomics and Human Factors and a member of the Institution of Engineering and Technology. He has been published in academic journals including "Nature". He has also helped organisations design new human-machine interfaces, such as the Adaptive Cruise Control system for Jaguar Cars.

Finding Nemo
Theatrical release poster
Directed by Andrew Stanton
Produced by Graham Walters
Screenplay by Andrew Stanton Bob Peterson David Reynolds
Story by Andrew Stanton
Starring Albert Brooks Ellen DeGeneres Alexander Gould Willem Dafoe
Music by Thomas Newman
Cinematography Sharon Calahan Jeremy Lasky
Edited by David Ian Salter
Production company Walt Disney Pictures Pixar Animation Studios
Distributed by Buena Vista Pictures Distribution
Release date May 30, 2003 (2003-05-30)
Running time 100 minutes
Country United States
Language English
Budget $94 million
Box office $940.3 million

Question: When was Neville A. Stanton's employer founded?
Thought: The employer of Neville A. Stanton is University of Southampton. The University of Southampton was founded in 1862. So the answer is: 1862.

> 👦 **User:**
> {{ retrieved_passages }}
> Question: {{ query }}
> Thought:

## C.2 PAGE NAVIGATION PROMPTS

**Page Navigation — Links Only**

**System prompt:**
You are a helpful assistant who can interact with a custom interface to expertly navigate information networks. The interface consists of a page viewer that shows you the current page contents with clickable links to related pages.

---

👦 **User:**
<start_page>
START_PAGE_TITLE: {{ start_page_title }}
START_PAGE_ID: {{ start_page_link_id }}
</start_page>
<target_page>
TARGET_PAGE_TITLE: {{ target_page_title }}
TARGET_PAGE_ID: {{ target_page_link_id }}
</target_page>

<instructions>
# Task Instructions
You need to navigate from the given START page to the TARGET page using the provided commands in as few steps as possible.
You may use any strategy, but your goal is to reach the exact page_id of the TARGET page, not just a similar title.
You have not reached the TARGET page unless the CURRENT_PAGE_ID matches the TARGET_PAGE_ID exactly even if the titles seem similar.
You succeed only when CURRENT_PAGE_ID == TARGET_PAGE_ID.

For example:
TARGET_PAGE_ID: Elon_Musk
CURRENT_PAGE_ID: Elon

These are different pages. Despite the similarity in names, you must land on the exact page ID to complete the task.

# Navigation Tips
- Use hub pages (countries, years, broad categories) to bridge between different topics
- Go broader before going narrower - find shared categories or themes
- Look for pages with many links when you need more options
- Think about what connects your start and target (time period, location, field, etc.)

IMPORTANT: This is an interactive process where you will think and issue ONE command via function calling, see its result, then think and issue your next command.
In each step, please output your thinking so that we can follow along.
Your thinking should be thorough and so it's fine if it's very long.
</instructions>
Begin!
{{ start_observation }}

---

**Page Navigation — Content + Links**

**System prompt:**
You are a helpful assistant who can interact with a custom interface to expertly navigate information networks. The interface consists of a page viewer that shows you the current page contents with clickable links to related pages. Links are displayed in markdown format as [entity](link).

---

👤 **User:**
<start_page>
START_PAGE_TITLE: {{ start_page_title }}
START_PAGE_ID: {{ start_page_link_id }}
START_PAGE_CONTENT:
{{ start_page_content }}
</start_page>
<target_page>
TARGET_PAGE_TITLE: {{ target_page_title }}
TARGET_PAGE_ID: {{ target_page_link_id }}
TARGET_PAGE_CONTENT:
{{ target_page_content }}
</target_page>

<instructions>
# Task Instructions
You need to navigate from the given START page to the TARGET page using the provided commands in as few steps as possible.
You may use any strategy, but your goal is to reach the exact page_id of the TARGET page, not just a similar title.
You have not reached the TARGET page unless the CURRENT_PAGE_ID matches the TARGET_PAGE_ID exactly even if the titles seem similar.
You succeed only when CURRENT_PAGE_ID == TARGET_PAGE_ID.

For example:
TARGET_PAGE_ID: Elon_Musk
CURRENT_PAGE_ID: Elon

These are different pages. Despite the similarity in names, you must land on the exact page ID to complete the task.

# Navigation Tips
- Use hub pages (countries, years, broad categories) to bridge between different topics
- Go broader before going narrower - find shared categories or themes
- Look for pages with many links when you need more options
- Think about what connects your start and target (time period, location, field, etc.)

IMPORTANT: This is an interactive process where you will think and issue ONE command via function calling, see its result, then think and issue your next command.
In each step, please output your thinking so that we can follow along.
Your thinking should be thorough and so it's fine if it's very long.
</instructions>

Begin!
{{ start_observation }}

---

**Click Page Tool Definition**

```
def click_link_to_page(
    self,
    page_id: Annotated[
        str,
        "The ID of the page to navigate to. The ID must be a link on the LATEST CURRENT
            PAGE.",
    ],
```

```
):
    """
    From the links on the current page, click on a link to the next page.
    """
```

**Backtrack to Page in History Tool Definition**

```
def backtrack_to_page_in_history(
    self,
    page_id: Annotated[
        str,
        "The ID of the page to navigate to. You may only click on links on the in the
            NAVIGATION_HISTORY.",
    ],
):
    """
    Backtrack to a specific page in the navigation history.
    """
```

# D  ADDITIONAL EXPERIMENT TABLES

Table 4 shows the results aggregated across all task instances for multi-hop QA. Table 5 shows the results aggregated across all task instances for page navigation.

| Model | Baseline | RM | | SM | | KA |
| | | RM (F1) | RM (R@5) | SM (F1) | SM (R@5) | KA (F1) |
| --- | --- | --- | --- | --- | --- | --- |
| GPT-5-mini | Closed-book | 21.6 [19.5, 23.7] | – | 0.2 [0.1, 0.4] | – | 21.4 |
| | Reading Comp | 88.1 [86.4, 89.8] | – | 90.1 [88.5, 91.6] | – | -2.0 |
| | One-step RAG | 49.8 [47.1, 52.4] | 56.1 | 24.4 [22.0, 26.8] | 45.0 | 25.4 |
| | IRCoT + RAG | 54.3 [51.7, 56.9] | 58.9 | 38.1 [35.4, 40.7] | 52.2 | 16.2 |
| Gemini-2.0-Flash | Closed-book | 19.4 [17.3, 21.4] | – | 0.6 [0.3, 0.9] | – | 18.8 |
| | Reading Comp | 75.4 [73.0, 77.8] | – | 80.3 [78.1, 82.4] | – | -4.9 |
| | One-step RAG | 37.3 [34.7, 39.9] | 56.1 | 17.2 [15.2, 19.3] | 45.1 | 20.1 |
| | IRCoT + RAG | 46.8 [44.1, 49.4] | 60.6 | 38.3 [35.5, 40.9] | 57.5 | 8.5 |
| gpt-oss-20b | Closed-book | 17.1 [13.3, 20.2] | – | 1.6 [0.0, 1.9] | – | 15.5 |
| | Reading Comp | 66.4 [62.4, 69.6] | – | 76.7 [72.3, 80.0] | – | -10.3 |
| | One-step RAG | 41.0 [38.0, 44.1] | 55.9 | 20.2 [17.6, 22.8] | 44.6 | 20.8 |
| | IRCoT + RAG | 30.7 [27.5, 33.9] | 45.2 | 18.4 [15.7, 21.2] | 38.5 | 12.3 |
| gpt-oss-120b | Closed-book | 21.6 [19.6, 23.7] | – | 1.8 [1.4, 2.3] | – | 19.8 |
| | Reading Comp | 73.6 [71.3, 75.8] | – | 82.8 [80.8, 84.8] | – | -9.2 |
| | One-step RAG | 43.5 [41.0, 46.1] | 55.9 | 22.1 [19.8, 24.4] | 44.6 | 21.5 |
| | IRCoT + RAG | 38.3 [35.6, 41.0] | 43.5 | 23.6 [20.5, 26.8] | 38.2 | 14.7 |

Table 4: **Multi-hop QA Performance on SYNTHWORLD-RM/SM.** Metrics are F1 scores (with 95% confidence intervals) for answer correctness and Recall@5 (R@5) for retrieval. The rightmost column reports the knowledge advantage gap (KA).

| Model | Environment | RM | SM | KA |
|---|---|---|---|---|
| GPT-5-mini | Links Only | 50.8 [47.7, 53.9] | 19.8 [17.4, 22.4] | 31.0 |
| | Content + Links | 52.3 [49.2, 55.4] | 30.6 [27.8, 33.5] | 21.7 |
| Gemini-2.0-Flash | Links Only | 36.1 [33.2, 39.1] | 15.6 [13.5, 18.0] | 20.5 |
| | Content + Links | 41.5 [38.5, 44.6] | 28.0 [25.3, 30.9] | 13.5 |
| gpt-oss-20b | Links Only | 28.6 [25.9, 31.5] | 13.2 [11.2, 15.4] | 15.4 |
| | Content + Links | 31.5 [28.7, 34.4] | 22.4 [19.9, 25.1] | 9.1 |
| gpt-oss-120b | Links Only | 39.9 [36.9, 43.0] | 16.2 [14.0, 18.6] | 23.7 |
| | Content + Links | 45.6 [42.5, 48.7] | 30.3 [27.5, 33.2] | 15.3 |
| Kimi-K2-Instruct | Links Only | 45.3 [42.2, 48.4] | 17.4 [15.2, 19.9] | 27.9 |
| | Content + Links | 49.4 [46.3, 52.5] | 31.6 [28.8, 34.5] | 17.8 |
| Kimi-K2-Thinking | Links Only | 43.6 [40.6, 46.7] | 16.2 [14.0, 18.6] | 27.4 |
| | Content + Links | 45.4 [42.3, 48.5] | 28.5 [25.8, 31.4] | 16.9 |

Table 5: **Navigation Success Rate on SYNTHWORLD-RM/SM.** Success rates are reported with 95% confidence intervals. The rightmost column reports the knowledge advantage gap (KA).

