# OpenReview forum: "SynthWorlds: Controlled Parallel Worlds for Disentangling Reasoning and Knowledge in Language Models"
_ICLR.cc/2026/Conference — ICLR 2026 Poster_

### Official Review · Reviewer_Ra5g · 2025-10-16

**Soundness:** 3
**Presentation:** 3
**Contribution:** 3
**Rating:** 6
**Confidence:** 3

**Summary:**

This paper investigates the real capability of reasoning of LLMs when isolating parametric knowledge. They design a simulated world from the Wiki so that both simulated world and real world share the same degree of difficulty. After that, they conduct detailed experiments and analyze the effect of reasoning and knowledge.

**Strengths:**

- This paper provides a detailed construction pipeline for a simulated world from the real world, including entity selection, question construction, and human annotation.
- This paper rigorously analyzes the effect of reasoning disentangled from parametric knowledge, using commonsense/multi-hop QA and page navigation.

**Weaknesses:**

- All the experiments are based on closed-source models. Some open-sourced models should be included, e.g., DeepSeek-V3.1 and K2, and various model sizes should be considered for generability, e.g., Qwen3 series.

**Questions:**

- For page navigation, is the content simulated or from the real world? I wonder if you could report the difference between the minimum steps needed to reach the final page and the steps LLMs take to reach the final page to see whether there is a potential shortcut.
- Could you explain why the F1-Score of Multi-hop QA of SM is higher in reading comprehension, since it is counterfactual (maybe) and should not be higher than RM.
- Coule you train an embedding model from scratch using you real-world corpora and simulated-world corpora separately to ablate the effect of LLM retrievals since they are constructed from the same source?

---

> ### Author Response · Authors · 2025-12-02
> **Response to Weakness**
>
> Thank you for recognizing our detailed construction pipeline and rigorous analysis of reasoning disentangled from parametric knowledge. We address your concerns below.
>
> > W1: Should include open-source models (DeepSeek-V3.1, K2) and various sizes (Qwen3 series) for generalizability.
>
> Excellent point. We have now evaluated 4 additional open-source models to demonstrate generalizability.  These were chosen due to their recency, and being able to make direct comparisons of model size as well as instruct vs thinking post-trained models.
>
> - **Scaling analysis**: `gpt-oss-20b` and `gpt-oss-120b` (understanding scale effects on KA) on multi-hop QA and page navigation
> - **Post-training effects**: `kimi-k2-instruct` and `kimi-k2-thinking` (impact of reasoning-focused training) on page navigation (results still running for QA)
>
> These experiments total an additional **40K+ evaluations** across SM and RM tasks, models, and baseline conditions.
> Due to budget constraints and the unlikely additional insight given the cost, we chose not to include Qwen3 and DeepSeek models.
>
> ---
>
> **QA: Closed-book, One-step RAG, & IRCoT + RAG**
>
> | Model | CB RM | CB SM | CB KA | RAG RM | RAG SM | RAG KA | IRCoT RM | IRCoT SM | IRCoT KA | KA Δ (RAG) | KA Δ (IRCoT) |
> |-------|:-----:|:-----:|:-----:|:------:|:------:|:------:|:--------:|:--------:|:--------:|:----------:|:------------:|
> | GPT-5-mini | 21.6 | 0.2 | 21.4 | 49.8 | 24.4 | 25.4 | 54.3 | 38.1 | 16.2 | +4.0 | -5.2 |
> | Gemini-2.0-Flash | 19.4 | 0.6 | 18.8 | 37.3 | 17.2 | 20.1 | 46.8 | 38.3 | 8.5 | +1.3 | -10.3 |
> | gpt-oss-20b | 17.1 | 1.6 | 15.5 | 41.0 | 20.2 | 20.8 | 30.7 | 18.4 | 12.3 | +5.3 | -3.2 |
> | gpt-oss-120b | 21.6 | 1.8 | 19.8 | 43.5 | 22.1 | 21.5 | 38.3 | 23.6 | 14.7 | +1.7 | -5.1 |
>
> *CB = Closed-book, RAG = One-step RAG, IRCoT = IRCoT + RAG*
> *KA Δ (RAG) = RAG KA - Closed-book KA*
> *KA Δ (IRCoT) = IRCoT KA - Closed-book KA*
>
> ---
>
> **Navigation: Links Only & Content + Links**
>
> | Model | Links Only RM | Links Only SM | Links Only KA | Content+Links RM | Content+Links SM | Content+Links KA | KA Δ |
> |-------|:-------------:|:-------------:|:-------------:|:----------------:|:----------------:|:----------------:|:----:|
> | GPT-5-mini | 50.8 | 19.8 | 31.0 | 52.3 | 30.6 | 21.7 | -9.3 |
> | Gemini-2.0-Flash | 36.1 | 15.6 | 20.5 | 41.5 | 28.0 | 13.5 | -7.0 |
> | gpt-oss-20b | 28.6 | 13.2 | 15.4 | 31.5 | 22.4 | 9.1 | -6.3 |
> | gpt-oss-120b | 39.9 | 16.2 | 23.7 | 45.6 | 30.3 | 15.3 | -8.4 |
> | Kimi-K2-Instruct | 45.3 | 17.4 | 27.9 | 49.4 | 31.6 | 17.8 | -10.1 |
> | Kimi-K2-Thinking | 43.6 | 16.2 | 27.4 | 45.4 | 28.5 | 16.9 | -10.5 |
>
> *KA Δ = Content+Links KA - Links Only KA*
>
> ---
>
> These results demonstrate that **knowledge shortcuts are robust** across:
> - **6 diverse models** (GPT-5-mini, Gemini-2.0-Flash, gpt-oss-20b/120b, Kimi-K2-Instruct/Thinking)
> - **Multiple task types** (QA, Navigation)
> - **Information availability** (Links Only, Content+Links for navigation; Closed-book, RAG, IRCoT for QA)
>
> Key findings:
> 1. **Knowledge advantage persists** with retrieval enabled (RAG/IRCoT) and when full content is provided
> 2. **The gap is relatively stable** even with RAG but closes with IRCoT (though performance actually drops for gpt-oss due to poor instruction following of the IRCoT format)
> 3. **Larger models and stronger performing models exhibit larger knowledge shortcuts** in navigation (e.g., comparing gpt-oss-20b vs. gpt-oss-120b or gemini-2.0-flash vs kimi models)
> 4. **Navigation content reduces but doesn't eliminate KA**
> 5. **Instruct/Thinking do not impact page navigation** between Kimi-K2-Instruct and Kimi-K2-Thinking
>
> This confirms models systematically exploit memorized knowledge as shortcuts across architectures and scales. These results are included in the revision in Section 5 and Table 4 and 5 in the Appendix.

---

> > ### Author Response · Authors · 2025-12-02
> > **Response to Questions**
> >
> > > Q1: For page navigation, is the content simulated or from the real world? I wonder if you could report the difference between the minimum steps needed to reach the final page…
> >
> > The content is LM-generated but based on real-world or synthetic mapped entities.
> > We can report shortest paths between pages, but that is not necessarily the most informative, as even humans playing this game would not necessarily be able to reproduce the shortest path. We believe that quantifying the difficulty in expected random walk length is more informative of the difficulty of the navigation task. Generally, the models will always take longer than the shortest path, though it is unrealistic to expect shortest path performance.
> >
> >
> > > Q2: Could you explain why the F1-score of Multi-hop QA of SM is higher in reading comprehension, since it is counterfactual (maybe) and should not be higher than RM.
> >
> > Great observation! This is actually an important finding about parametric knowledge interference (highlighted in line 409-410). When models have access to gold documents (RC setting), parametric knowledge can lead models astray rather than help. Here are a few examples:
> > …
> > > Q3: Could you train embedding models separately on RM and SM corpora to ablate LM retrieval effects?
> >
> > Good question. For this work, the focus is on the disentangling of reasoning in LM systems in general, and assessing how existing LM systems (with existing retrievers) handle novel environments. Training embedding models for each new SM corpora may not be the most efficient strategy to incorporate additional knowledge. However, this is, of course, meaningful and can be studied using the datasets we developed using our framework.
> >
> > We note that in the HippoRAG2 system used in our experiments, the index is built (using an off-the-shelf LM) separately for the SM and RM corpora. This is why we observe retrieval quality differences (Table 4).

---

### Official Review · Reviewer_HETT · 2025-10-19

**Soundness:** 3
**Presentation:** 3
**Contribution:** 2
**Rating:** 4
**Confidence:** 4

**Summary:**

SynthWorlds proposes a fully automatic, scalable framework to disentangle language-model reasoning from memorized factual knowledge by constructing parallel corpora and tasks in two mirrored worlds—real-mapped (RM) and synth-mapped (SM)—and measuring a knowledge advantage (KA) gap, i.e., the performance difference between RM and SM conditions.

**Strengths:**

1. The paper offers a novel and well-motivated formulation of disentangling reasoning from memorized factual knowledge by constructing paired “real-mapped” and “synth-mapped” worlds with matched structure, and by introducing a formal knowledge-advantage metric to quantify the contribution of parametric knowledge.
2. The data-generation pipeline is fully automated; difficulty is explicitly controlled; the evaluations span parallel multi-hop QA and page navigation tasks; and the comparisons cover closed-book, single-step RAG, and iterative reasoning–retrieval (IRCoT + RAG) across multiple models with appropriate metrics, yielding credible, interpretable results.

**Weaknesses:**

1. Conclusions are based on two property models; it is unclear how KA scales with capacity or how post-training techniques affect KA.
2. The corpus and questions derived largely from one source (Wikidata) may yield relation distributions and writing style that favor certain generalization paths, which may disrupt the evaluation.
3. Although the benchmark proposed in the paper can quantify KA, the quantification results do not seem surprising. Moreover, it is not yet shown that improvements on SynthWorlds translate to gains on standard web or scientific tasks.

**Questions:**

1. Do the findings persist across model sizes within a family? Please discuss whether scaling reduces the gap via better retrieval or stronger reasoning.
2. Can you include at least one additional domain (e.g., biomedical/software) and report the results?
3. What are the dominant RM vs. SM failure modes?

---

> ### Author Response · Authors · 2025-12-02
> **Response to W1 and Q1**
>
> Thank you for recognizing our "novel and well-motivated formulation" and the comprehensive evaluation design. We address your concerns below and believe the revisions will strengthen the paper's contribution.
>
>
> > W1 and Q1:  Conclusions based on two proprietary models; unclear how KA scales with capacity or post-training…Do the findings persist across model sizes within a family? Please discuss whether scaling reduces the gap via better retrieval or stronger reasoning.
>
> Excellent point. We have now evaluated **4 additional open-source models**, which were chosen due to their recency, and being able to make direct comparisons of model size as well as instruct vs thinking post-trained models:
>
> - **Scaling analysis**: `gpt-oss-20b` and `gpt-oss-120b` (understanding scale effects on KA) on multi-hop QA and page navigation
> - **Post-training effects**: `kimi-k2-instruct` and `kimi-k2-thinking` (impact of reasoning-focused training) on page navigation (results still running for QA)
>
> These experiments total an additional **40K+ evaluations** across SM and RM tasks, models, and baseline conditions.
>
> ---
>
> **QA: Closed-book, One-step RAG, & IRCoT + RAG**
>
> | Model | CB RM | CB SM | CB KA | RAG RM | RAG SM | RAG KA | IRCoT RM | IRCoT SM | IRCoT KA | KA Δ (RAG) | KA Δ (IRCoT) |
> |-------|:-----:|:-----:|:-----:|:------:|:------:|:------:|:--------:|:--------:|:--------:|:----------:|:------------:|
> | GPT-5-mini | 21.6 | 0.2 | 21.4 | 49.8 | 24.4 | 25.4 | 54.3 | 38.1 | 16.2 | +4.0 | -5.2 |
> | Gemini-2.0-Flash | 19.4 | 0.6 | 18.8 | 37.3 | 17.2 | 20.1 | 46.8 | 38.3 | 8.5 | +1.3 | -10.3 |
> | gpt-oss-20b | 17.1 | 1.6 | 15.5 | 41.0 | 20.2 | 20.8 | 30.7 | 18.4 | 12.3 | +5.3 | -3.2 |
> | gpt-oss-120b | 21.6 | 1.8 | 19.8 | 43.5 | 22.1 | 21.5 | 38.3 | 23.6 | 14.7 | +1.7 | -5.1 |
>
> *CB = Closed-book, RAG = One-step RAG, IRCoT = IRCoT + RAG*
> *KA Δ (RAG) = RAG KA - Closed-book KA*
> *KA Δ (IRCoT) = IRCoT KA - Closed-book KA*
>
> ---
>
> **Navigation: Links Only & Content + Links**
>
> | Model | Links Only RM | Links Only SM | Links Only KA | Content+Links RM | Content+Links SM | Content+Links KA | KA Δ |
> |-------|:-------------:|:-------------:|:-------------:|:----------------:|:----------------:|:----------------:|:----:|
> | GPT-5-mini | 50.8 | 19.8 | 31.0 | 52.3 | 30.6 | 21.7 | -9.3 |
> | Gemini-2.0-Flash | 36.1 | 15.6 | 20.5 | 41.5 | 28.0 | 13.5 | -7.0 |
> | gpt-oss-20b | 28.6 | 13.2 | 15.4 | 31.5 | 22.4 | 9.1 | -6.3 |
> | gpt-oss-120b | 39.9 | 16.2 | 23.7 | 45.6 | 30.3 | 15.3 | -8.4 |
> | Kimi-K2-Instruct | 45.3 | 17.4 | 27.9 | 49.4 | 31.6 | 17.8 | -10.1 |
> | Kimi-K2-Thinking | 43.6 | 16.2 | 27.4 | 45.4 | 28.5 | 16.9 | -10.5 |
>
> *KA Δ = Content+Links KA - Links Only KA*
>
> ---
>
> These results demonstrate that **knowledge shortcuts are robust** across:
> - **6 diverse models** (GPT-5-mini, Gemini-2.0-Flash, gpt-oss-20b/120b, Kimi-K2-Instruct/Thinking)
> - **Multiple task types** (QA, Navigation)
> - **Information availability** (Links Only, Content+Links for navigation; Closed-book, RAG, IRCoT for QA)
>
> Key findings:
> 1. **Knowledge advantage persists** with retrieval enabled (RAG/IRCoT) and when full content is provided
> 2. **The gap is relatively stable** even with RAG but closes with IRCoT (though performance actually drops for gpt-oss due to poor instruction following of the IRCoT format)
> 3. **Larger models and stronger performing models exhibit larger knowledge shortcuts** in navigation (e.g., comparing gpt-oss-20b vs. gpt-oss-120b or gemini-2.0-flash vs kimi models)
> 4. **Navigation content reduces but doesn't eliminate KA**
> 5. **Instruct/Thinking do not impact page navigation** between Kimi-K2-Instruct and Kimi-K2-Thinking
>
> This confirms models systematically exploit memorized knowledge as shortcuts across architectures and scales. These results are included in the revision in Section 5 and Table 4 and 5 in the Appendix.

---

> > ### Author Response · Authors · 2025-12-02
> > **Response to W2, W3 and Q2**
> >
> > > W2: Corpus from Wikidata may favor certain generalization paths, disrupting evaluation
> >
> > Our goal is to evaluate reasoning over realistic knowledge structures. Generating novel text with verifiable properties we care about (e.g., entity and relation types) is hard without grounding in prior artifacts. Likewise, we consider it an important advantage that this grounding in real-world artifacts allows us to track and evaluate the properties of our dataset during and after generation (e.g., Fig. 6, 7, 8).
> >
> > Wikidata provides both structure and complex relationships that mirror real-world knowledge relationships and reasoning tasks (100M+ entities, 10K+ relation types, sparse graphs, and heavy-tailed distributions) while enabling the automated, controlled perturbations necessary for our framework. As such, it is a natural fit for this work.
> >
> > **Deriving corpora directly from web text would sacrifice this control:** we couldn't systematically select which entities or relationships to manipulate, nor validate that generated facts remain coherent and consistent. Nevertheless, the framework could be extensible to not just Wikidata, and one could imagine a corpus built from mixing multiple knowledge graphs or from other domains (see our discussion on other domains below).
> >
> > The core idea, sampling a subgraph, renaming entities, and constructing parallel corpora and tasks, generalizes to any knowledge graph and other domains. We discuss this potential as an opportunity for future work in Section 6.
> >
> >
> > > W3: Quantification results don't seem surprising. Not shown that improvements on SynthWorlds translate to gains on standard tasks
> >
> > While prior work measures reasoning on standalone datasets, through parallel datasets and tasks, we quantify the performance difference when knowledge is obstructed. We also measure the extent to which this gap can be closed with current knowledge augmentation approaches (e.g., RAG). **Defining conceptually, operationalizing concretely, and experimentally quantifying this Knowledge Gap is something that has not existed.**
> >
> > **Key findings include:**
> > - The KA gap persists even with iterative retrieval+reasoning (IRCoT+RAG)
> > - Retrieval itself shows gaps when using HippoRAG2 (LM-based indexing)
> > - The size of this Knowledge Gap magnitude varies by reasoning motif, providing actionable insights
> >
> > Furthermore, while not our primary goal, performance on SynthWorld QA shows positive correlation with the Artificial Analysis Intelligence Index (e.g., on RM for QA, pearson correlation is between 0.25 and 0.83 for closed-book, reading comprehension, and one-step RAG baselines (failure modes on IRCoT + RAG discussed below). This demonstrates that our tasks capture broader reasoning capabilities, while the parallel exactly mirrored feature enables precise measurement of the KA gap across baselines.
> >
> >
> >
> > > Q2: Can you include at least one additional domain (e.g., biomedical/software) and report the results?
> >
> > Our work aims to demonstrate the framework with two tasks and a case study in a broad domain. **We wanted to prioritize one high-quality eval first** (experiments on thousands of instances, across multiple RAG and agentic baselines, and 6 models), rather than spend this compute budget on creating meaningful evaluations and experiments in another domain.
> >
> > Given the page limit, we are also concerned that describing another domain and case study would require space that would take away from the existing work that multiple reviewers have described positively. We already have substantial appendices to capture full details of the Wikidata-based pipeline and want to ensure that our pipeline is easily reproducible by others. Adding more domains would severely challenge readability and reproducibility, given the limited space available. **We also make no claims that we have proved this generalizability already.** However, we do agree that this is an important area for future work.
> >
> > **How this might work** is either starting with a knowledge graph in a particular domain, or mining knowledge graph relationships from text in that domain (e.g., GitHub and Stack Overflow) using modern KG extraction approaches [1,2]. We then thoughtfully determine the knowledge we want to perturb and do this consistently, following the SynthWorld framework. For example, for math, we could think about creating parallel worlds with different notation systems: RM: $x,y,f(x)$; SM: $α,β,φ(α)$, etc. Similarly, for code generation, we can consistently rename entire libraries (e.g., `numpy/pandas`) and function calls for the SM variant. We include this discussion as future work in Section 6.

---

> > > ### Author Response · Authors · 2025-12-02
> > > **Response to Q3**
> > >
> > > > Q3: What are the dominant RM vs. SM failure modes?
> > >
> > > By generating our mirrored datasets, we can precisely study model behavior in controlled conditions.
> > > We conducted a detailed qualitative analysis of GPT-5-mini across three task types. Our findings reveal distinct failure modes that validate SynthWorlds' utility for measuring and comparing model behavior.
> > >
> > > **For reading comprehension**, we a common RM failure modes in knowledge leakage as the model uses parametric knowledge instead of reading the passage
> > >
> > > | Instance | RM Query | RM Prediction | Ground Truth | Why RM Failed |
> > > |----------|----------|---------------|--------------|---------------|
> > > | 281 | "In which country is the place where the person who influenced Wifredo Lam worked?" | "France" ❌ | "Kingdom of Sardinia" | Model knows Henri Matisse worked in Nice, France but the passage describes the *historical* context where Nice was in the Kingdom of Sardinia |
> > > | 440 | "Who is the chairperson of the political party Mikhail Lavrentyev belongs to..." | "Nikita Khrushchev" ❌ | "Joseph Stalin" | Model confuses Soviet-era leaders from parametric memory |
> > >
> > > In contrast, SM forces faithful passage reading as the model must extract the answer from context, yielding correct predictions.
> > >
> > > ---
> > >
> > > **In One-step RAG**,  RM outperforms SM with the common failure mode being poorer retrieval.
> > >
> > > | Instance | RM Recall@5 | SM Recall@5 | RM Pred | SM Pred |
> > > |----------|-------------|-------------|---------|---------|
> > > | 1 | 1.0 | 0.5 | "1825" ✓ | "not stated in the provided text" ❌ |
> > > | 2 | 1.0 | 0.5 | "salzburg" ✓ | "not specified" ❌ |
> > > | 5 | 1.0 | 0.5 | "1896-07-28" ✓ | "5 june 1876" ❌ |
> > > | 8 | 1.0 | 0.5 | "hungarian" ✓ | "not stated" ❌ |
> > >
> > > When SM retrieval succeeds (Recall@5 = 1.0), answer accuracy becomes comparable, indicating the bottleneck is retrieval, not reasoning. (discussed in line 426)
> > >
> > > ---
> > >
> > > **For navigation,** a common SM failure mode is a lack of navigation cues.
> > >
> > > **Illustrative Example (Instance 55):**
> > > | | RM | SM |
> > > |--|----|----|
> > > | **Start** | "Shore Leave" (Star Trek episode) | "The Silent Guardian" (Eldoria Chronicles episode) |
> > > | **Target** | "German" | "Malquari" |
> > > | **Steps** | 2 ✓ | 30 (max steps reached) ❌ |
> > > | **Agent Reasoning** | "I will click Gert_Günther_Hoffmann because that person is **likely German**" | "Kaldor Byrnes sounds like a distinctive name that might be connected to regions..." |
> > >
> > > In RM, the model leverages world knowledge that "Gert Günther Hoffmann" is a German name, enabling a direct 2-hop path. In SM, the synthetic name "Kaldor Byrnes" provides no such signal (the agent explores randomly for 30 steps without finding "Malquari").
> > >
> > > These findings validate SynthWorlds' core contribution: parallel worlds isolate model capabilities by eliminating memorized parametric knowledge in the SM world.

---

### Official Review · Reviewer_2dyf · 2025-10-20

**Soundness:** 3
**Presentation:** 4
**Contribution:** 3
**Rating:** 6
**Confidence:** 4

**Summary:**

SYNTHWORLDS introduces a novel framework to disentangle reasoning from factual knowledge in language models by constructing parallel corpora: a real-mapped world with familiar entities and a synthetic-mapped world with renamed entities, preserving identical reasoning structures while obscuring parametric knowledge. Through case studies on multi-hop question answering and page navigation, experiments reveal a persistent knowledge advantage gap, where LMs like GPT-5-mini and Gemini-2.0-Flash perform better in RM settings even with retrieval augmentation, highlighting reliance on memorized knowledge. This scalable, automated approach enables controlled evaluation, offering datasets and tasks that facilitate precise analysis of reasoning abilities, with contributions including a generation pipeline, public corpora, and empirical insights into LM limitations in novel environments.

**Strengths:**

- SYNTHWORLDS uniquely separates reasoning complexity from parametric knowledge through parallel corpora.
- The framework is fully automatic and scalable, leveraging knowledge graphs (e.g., Wikidata) to generate large, interconnected corpora without manual curation.

**Weaknesses:**

- This paper proposes the challenges in distinguishing reasoning from reciting for controlled evaluation. However, we don't know whether the paper really solve this problem. I mean, if your scores can precisely reflect the real reasoning abilities of LLMs, then you should observe a correlation between human preference (e.g. LM Arena Rankings on Reasoning) and your scores.
- This paper mentions two kinds previous approaches on controlled evaluation: (1) curation of “clean” evaluation sets and (2) synthetic
dataset generation, but the authors did not compare their method with these methods in experiments. I know you may argue you did not know how to measure the effectiveness of evaluation methodologies on controlled evaluation. You may refer to this work's settings for comparison experiments [1].
- Typo: Line 102: " (§5).Across" has no space between ")" and "Across".


[1]  Establishing Trustworthy LLM Evaluation via Shortcut Neuron Analysis. In Proceedings of the 63rd Annual Meeting of the Association for Computational Linguistics (Volume 1: Long Papers), pages 3809–3822, Vienna, Austria. Association for Computational Linguistics.

**Questions:**

- The paper defines the Knowledge Advantage Gap as a core metric. To what extent can this metric be generalized to other domains beyond the structured knowledge of Wikidata (e.g., mathematical reasoning or code generation)?
- If the framework and its generated datasets become widely adopted, what prevents future LMs from being trained on generations of SYNTHWORLDS-style corpora?

---

> ### Author Response · Authors · 2025-12-02
> **Response to Weaknesses**
>
> Thank you for recognizing SynthWorld’s purposeful design in separating reasoning complexity from parametric knowledge, and for noting our framework's scalability and automation. We address your concerns below.
>
> > W1: If scores reflect real reasoning abilities, they should correlate with LM Arena rankings.
>
> We highlight that distinguishing reasoning from memory is our primary goal, in which it is important to first understand the performance gap in RM and SM worlds, and then the raw performance of reasoning.
>
> Our central idea and contribution is to capture the KA gap for reasonably constructed and mirrored tasks in RM and SM worlds, which helps us understand to what extent memory can impact benchmark performance. SynthWorlds also presents novel and continual opportunities to study how LM systems acquire novel information in the absence of knowledge. This is different from the goal/measurement provided by LM Arena, i.e., human preferences on tasks. We see that in our tasks, the KA gap can be up to 31.0 (Table 5), which is something existing approaches do not capture.
>
> Additionally, our results align with existing reasoning benchmarks. while not our primary goal, performance on SynthWorld QA shows positive correlation with the Artificial Analysis Intelligence Index (e.g., on RM for QA, pearson correlation is between 0.25 and 0.83 for closed-book, reading comprehension, and one-step RAG baselines).  However, we highlight again that our goal is *not to reproduce findings* from human-collected reasoning datasets in our synthetic and scalable datasets, but to introduce a novel evaluation framework in which the model is **unfamiliar with its environment and must integrate novel knowledge**, and then **compare** this performance to an environment where the model has potentially seen the facts before, to then **quantify the performance gap**.
>
> >W2: Mentions clean evaluation sets and synthetic datasets, but no experimental comparison.
>
> We thank you for the reference to [1]. We clarify an important distinction: [1] addresses data contamination detection using neuron analysis, which requires white-box model access. While we could theoretically apply their method to measure performance deltas on our dataset (or other datasets), doing so would require model weights. We view our approach as complementary to these efforts, as our framework provides a key contribution in a **novel understanding** of the **knowledge advantage gap for black-box models** through parallel datasets and mirrored tasks.
>
> Regarding our discussion of clean datasets and synthetic data generation, we mentioned these approaches as motivation for our work, highlighting their limitations. Clean datasets require significant manual curation effort and can still be memorized over time. Many synthetic data generation methods modify existing data. Our parallel-world framework naturally addresses these drawbacks by being fully automated and generating novel data. The novelty of our generated dataset is validated both during our pipeline (line 1064) and in the 0% performance on closed-book QA (Fig. 4).
>
> While we could measure the "contamination" of existing datasets (both human and/or synthetically constructed), there are adequate works and examples that show this already (see our intro), including [1]. In addition, by generating both "novel" and "seen" data at the corpus level that is exactly mirrored, we can precisely study model behavior (i.e., what knowledge they learn, assume, and connect) in controlled conditions. How do models approach a task when they must integrate novel information vs. when they have some knowledge of the entities already? This is something *unique to our work* that clean datasets, existing synthetic datasets, arena-type evaluations, and [1] *do not address*. We add this point to our related work.
>
> If you believe additional experiments comparing evaluation methodologies would strengthen the work, we would appreciate specific guidance on what comparisons would address your concern.

---

> > ### Author Response · Authors · 2025-12-02
> > **Response to Typo Suggestion and Questions**
> >
> > > T1: Line 102: " (§5). Across" has no space between ")" and "Across".
> >
> > Thanks for the suggestion. We have fixed the typo.
> >
> > > Q1: To what extent can this (KA) metric be generalized to other domains (e.g., mathematical reasoning or code generation)?
> >
> > The KA metric essentially needs parallel data in which one set is grounded in real-world knowledge, while the other is not, while keeping domain-general knowledge intact. The idea of perturbing certain properties in a certain domain to achieve this can be generalized, but requires thoughtful design that likely will depend on specific domains. For example, for math, we could think about creating parallel worlds with different notation systems: RM: $x,y,f(x)$; SM: $α,β,φ(α)$, etc. Similarly, for code generation, we can consistently rename entire libraries (e.g., `numpy/pandas`) and function calls for the SM variant. We include this discussion in Section 6.
> >
> > > Q2: What prevents future LMs from being trained on generations of SYNTHWORLDS-style corpora?
> >
> > Thanks for highlighting this important discussion. This was a core consideration that led to the many points of stochasticity in SynthWorlds' pipeline. While LMs can train on specific SynthWorlds-variants, the combinatorial space of knowledge graphs, subgraphs induced from the knowledge graphs (i.e., specific nodes, relationships, node-types), and renames, and our novelty validation pipeline can ensure that we can generate evaluation data that is resistant to fact memorization (similar to Gong et al.)

---

### Official Review · Reviewer_raJh · 2025-10-31

**Soundness:** 4
**Presentation:** 4
**Contribution:** 3
**Rating:** 8
**Confidence:** 5

**Summary:**

The paper introduces SynthWorlds, a framework to create multi-hop questions and page navigation benchmarks from a knowledge graph e.g. WikiData and assess an LM’s memorized factual knowledge. SynthWorlds builds two parallel corpora on a knowledge graph, one with real-world entities and another with synthetically generated ones, to keep reasoning complexity identical while removing factual familiarity that an LM could utilize in the real-world setting. The paper measures the “knowledge advantage gap” i.e. the LM’s parametric knowledge memorized in its weights, demonstrating this in frontier LLMs like GPT-5-mini and Gemini-2.0-Flash. Even in RAG-based approaches, this gap can exist.

**Strengths:**

1. The introduction is extremely well-written, just exquisite writing. I especially liked lines 59 through 76.
2. For synthetic multi-hop QA data generation to evaluate LMs, the paper presents the right next step by building synthetic datasets on top of the real-world Wikipedia knowledge graph that captures the complex interconnectedness and messiness. I think this paper is exciting for the field of multi-hop reasoning evaluation.
3. SynthWorlds contains a set of reasoning motifs, e.g., constraints and joins, that are not just logical compositions. This makes the questions more realistic; however, the question difficulty is unclear. I guess this is a tradeoff: in contrast to PhantomWiki that had logical compositions only, the question difficulty (reasoning steps) was simply the sum of the number of hops.

**Weaknesses:**

1. The authors heavily talk about determining knowledge gaps as the main contribution, but mention task reasoning difficulty in contribution 1 and Sec 3 line 205 as a main contribution. It is clear how SynthWorlds is evaluating knowledge gaps, but not clear at all how to determine task reasoning difficulty. It would be good to discuss this, since it's a main contribution of the work.
2. A major limitation is that SynthWorlds requires a knowledge graph to exist for a document corpus (Wikidata graph in the case of Wikipedia). I would be curious about what the authors think could work when we only have access to Wikipedia but not Wikidata.
3. Because the questions and documents from the graph G_facts are LM generated, there is no clear way to automatically verify full correctness. And what if the questions contain parametric knowledge from the LM generating the questions? There is discussion in App B.2 about human validation; however, I'm interested in automatic validation techniques or correct-by-design question/document generation.
4. Since the SM is created once, future LLMs would memorize it once uploaded to the internet. Is there a clear pipeline for creating synthetic worlds from scratch and any knowledge graph? Several synthetic benchmarks create knowledge graphs from scratch (although not as realistic as WikiData), and LLM memorization is not helpful by design, e.g., PhantomWiki and GSM-Infinite.

**Questions:**

1. Why not generate multiple SMs for a given RM? And take the average performance with stderr?
2. Line 194: The augmented case is unclear. What do the authors mean by "providing the model with external knowledge acquisition and integration strategies"? Does baseline just mean CoT prompting with a few CoT examples, and augmented mean RAG prompting? If so, it would be better to just say that.
3. Line 194: What is "near random"? Since these questions are open-ended QA and not multiple-choice, shouldn't P_S^base be near 0? i.e., the LM can't answer those questions at all in the baseline setting?
4. Line 202: The example of "rivers remain river-like..." does not improve clarity. Is there a better concrete example?
5. Figure 4: I'm confused why the Recall@5 for RM and SM (blue and orange) are not about the same. This is measuring the fraction of times the correct document was retrieved from the corpus, right? Since this is offloaded to an embedding-based retriever, why is the recall higher for RM over SM?
6. How are the authors calculating error bars in Fig 4, 5, etc.?
7. Are questions in page navigation baselines also formed using reasoning motifs? Then in Figure 5 it would make sense to club results by reasoning motifs, like in Figure 4. See the other comment about improving Figure 5 axes.

### Typos and editorial suggestions
1. Line 65: It would be useful to add citations for synthetic data generation that use existing content directly and template-based evaluation. This would parallel the previous example with ToolQA and Zhang et al., Mirzadeh et al. work.
2. Line 89: "Surface-form-consistent transformations" is an uncommon term in the literature, so it would be best to paraphrase this in layman's terms, or define it explicitly here.
3. Figure 5: Instead of breaking up the plots by expected random walk distance, it would be good to do a line plot, with success rate on the y-axis, expected random walk distance on the x-axis, a blue line for RM and orange for SM. Then you could have 2 plots per model, one for Links and another for Cont+Links. It makes more sense to demonstrate success as a function of difficulty. The current plot highlights Links vs. Cont+Links more, which I don't think is the main point that the authors are driving home. The plot would be similar to that of PhantomWiki (Gong et al.).

---

> ### Author Response · Authors · 2025-12-02
> **Response to W1-W3**
>
> Thank you for the detailed read of our work and your thoughtful and constructive feedback. We are excited by your recognition of SynthWorlds as an exciting step for building synthetic evaluation data and our efforts to capture the complex interconnectedness and messiness of real-world corpora. We respond to your questions and concerns below:
> > W1: not clear at all how to determine task reasoning difficulty. It would be good to discuss this …
>
> Great point. Both tasks use controllable attributes as difficulty proxies:
>
> * **Page navigation**: Expected random walk distance—higher values require more decision-making at each step.
> * **Multi-hop QA**: Number of hops (or number of decomposed single-hop questions) and motif complexity (Table 3). In our dataset, we have motifs with 2 decomposed questions (A), 3 (B, C), or 4 (D, E, F). We can also consider motifs containing others as subgraphs are strictly harder (D>B>A, F>C, E>C), where G1 > G2 denotes motif G2 is a subgraph of motif G1, as it requires one additional hop. The order of motifs presented left to right in Fig. 4 aims to show the general difficulty trend.
>
> We make this clarification in Section 4.1.
>
> > W2: requires a knowledge graph to exist for a document corpus…[what] could work when we only have access to Wikipedia.
>
> For unstructured text like Wikipedia, we considered renaming entities directly on pages to generate synthetic text; however, because the exact phrases, sentence structures, and document layouts are likely in the training corpus [1,2], we aimed to remove these surface-form confounders entirely by regenerating text from structured facts. Our framework requires a high-quality KG to encode complex knowledge relationships and ensure factual consistency in synthesized text.
>
>
> However, we can always construct KGs from text. For Wikipedia-only settings, modern extraction methods make KG construction from text increasingly feasible [3, 4]. Once the KG is built with consistent, accurate facts, the SynthWorlds pipeline follows naturally. We discuss this extension in Section 6.
>
> [1] Sainz et al., "GoLLIE: Annotation Guidelines improve Zero-Shot Information-Extraction", ICLR 2024
>
> [2] Xu et al., "Large Language Models for Generative Information Extraction: A Survey", arXiv:2312.17617, 2024
>
> [3] Hartmann et al., "SoK: Memorization in General-Purpose Large Language Models", arXiv:2310.18362, 2023
>
> [4] Ippolito et al., "Preventing Generation of Verbatim Memorization in Language Models Gives a False Sense of Privacy", INLG 2023
>
> > W3: Because the questions and documents from the graph G_facts are LM-generated, there is no clear way to verify full correctness automatically.
>
> This is a thoughtful point. Ensuring quality and correctness automatically or by construction was something we wanted to keep throughout our approach. Thus, each step in the pipeline was **narrowly scoped**, such that the task is feasible for LMs to achieve this level of accuracy. Each step is also inspired by prior work (Hennigen et al. and Josifoski et al.), in which they evaluated and found similar steps feasible with LMs (which were weaker than today’s LMs) for generating text from structured facts. We note that these are relatively simple tasks in which we also included programmatic filters or checks for quality (line 1054). The only part of the pipeline that could use human validation was converting facts into documents, from which our human validation found no inconsistencies across 28 pages and ~800 facts (Appendix B.5).
>
> > W3: What if the questions contain parametric knowledge from the LM generating the questions?
>
>
> Questions are generated from fact triplets with entity names replaced by synthetic alternatives. We can make the fair assumption that these renamed entities do not leak parametric knowledge, as synthetic names are randomly sampled from a set of LM-generated names conditioned only on the original entity type. Crucially, there is no mention of the original entity itself. Since question generation receives only entity types (not real-world entity names or relationships), the questions are novel by construction. See lines 1940 and 2039 in the appendix for the relevant prompts.
>
>
> We also verify that the documents themselves do not leak parametric knowledge in the novelty validation step (i.e., ask whether the LM that generated the document recognizes the real-world entity given the synth-mapped document and knowledge that the document contains perturbed entity names). Details in line 1064 with the prompt in line 1782. This does not consider whether or guarantee that the document is novel for every LM, but it is a simple and effective method for ensuring novelty. The zero performance of models on SM questions further validates and reinforces that the questions are about unseen knowledge.

---

> > ### Author Response · Authors · 2025-12-02
> > **Response to W4 and Q1-6**
> >
> > > W4: Is there a clear pipeline for creating synthetic worlds from scratch, and any knowledge graph?
> >
> > Yes, this is exactly what our approach proposes. Appendix A highlights the high-level framework, and Appendix B shows how this is instantiated in Wikidata. We make this clearer in Section 3 . For Wikidata, our pipeline could be used as it is, and we can generate completely new datasets by changing the hyperparameters (i.e., include/exclude certain entity types, different seed nodes, different knowledge graph sampling procedures) to get different corpora, even as LMs train on the existing dataset. In addition, our renaming is stochastic (LMs generate different names for a given entity type, and names are sampled for a given entity of an entity type) enabling us to generate different corpora even given a single knowledge subgraph.
> >
> >
> > For any new knowledge graph, there may be slightly different metadata informing what entities should and should not be renamed; however, the high-level framework still applies. The core idea of sampling a subgraph of a knowledge graph, sampling new names to rename entities, and constructing corpora and tasks from the renamed facts can be generally applied to any knowledge graph. See our response to 2dyf for a discussion on how our framework can be applied in multiple other settings, which we also add to Section 6.
> >
> > > Q1: Why not generate multiple SMs for a given RM? And take the average performance with stderr?
> >
> > To ensure a fair comparison with difficulty precisely controlled for, the SM and RM tasks are exactly mirrored with the document and wording (only entity names perturbed). Multiple SMs would just mean multiple renamings of the same SM structure (e.g., we don’t rename Mumbai to X but to Y). Since one SM already obstructs parametric knowledge, multiple SMs for the same SM only show the variation of LM performance when something unseen is renamed to something else unseen. While doable, we did not expect significant variation based on our initial experiments. Therefore, we instead chose to spend valuable compute resources on a larger number of tasks (2.2K in total) for accurate error bounds and across more baselines and models.
> > > Q2: Line: 194: What do the authors mean by "providing the model with external knowledge acquisition and integration strategies"?
> >
> > As we envision LMs interacting in novel environments, we believe there could be more ways to acquire novel knowledge external to the LMs' parametric knowledge to help them perform their goal. In multihop QA, external knowledge acquisition is obtained through retrieving documents in the corpus (whether through one step or multiple steps in CoT). In page navigation, this was through an LM agent reading page content, which was added to its context window. There may be other ways an agent may explore, interact, and store knowledge when placed in a novel environment. While we wanted to be general, we appreciate your comments for clarity, and we make this clearer in the paper.
> >
> > > Q3: Line: 194 What is "near random"?
> >
> > Thanks for the clarification. What we mean is 0 performance for multihop QA or performance equivalent to random walks (this may mean a success rate higher than 0 on easy navigation questions). We have fixed this in the paper.
> >
> > > Q4: The example of "rivers remain river-like..." does not improve clarity. Is there a better concrete example?
> >
> > Thanks for the suggestion. We remove the river example with this: university names remain university-like (University of Toronto -> University of Grandvale, not → Grandvale Bank) and libraries remain library-like (Central Library → Oakwood Public Library, not → Central Stadium).
> >
> > > Q5: I'm confused why the Recall@5 for RM and SM
> >
> > For our RAG system, we use HippoRAG2, which uses an LM to index the documents and during online retrieval (discussed in line 430-431; also see Fig. 2 in Gutierrez et al.). Thus, given the different input questions and documents between SM and RM, the RAG system itself may perform differently on RM and SM retrieval. This is also why we said “knowledge acquisition and integration strategies” as even for RAG, there are different approaches that could lead to different performance on novel knowledge. To make this easier to understand, we have added an explanation in Sec 6 RQ2.
> >
> > > Q6: How are the authors calculating error bars?
> >
> > F1 error bars (Fig. 4) show 95% bootstrap confidence intervals. For each difficulty bucket, we resample with replacement 5,000 times, compute the mean F1 (or F1-gap) for each resample, and take the 2.5th and 97.5th percentiles as confidence bounds.
> > Success-rate error bars (Fig. 5) show 95% Wilson score confidence intervals, computed via `statsmodels.stats.proportion.proportion_confint` with `method='wilson'` in Python.
> > We now incorporate these details in Appendix C.

---

> > > ### Author Response · Authors · 2025-12-02
> > > **Response to Q7 and Typo suggestions 1-3**
> > >
> > > > Q7: Are questions in page navigation baselines also formed using reasoning motifs?
> > >
> > > Page navigation questions are not formed using reasoning motifs. At each step during navigation, the current page fans out into many plausible next pages, so there is no single, traceable reasoning path. To operationalize difficulty, we instead use the expected random-walk length, i.e., the average number of steps a random walker would need to reach the target page. This captures proximity, branching structure, and the presence of multiple paths, similar to PageRank and other random-walk-based approaches. We then bucket questions by this walk length to provide a consistent, graph-based measure of difficulty.
> > >
> > > > T1: Line 65: ...useful to add citations for synthetic data generation that use existing content directly and template-based evaluation
> > >
> > > Great suggestion, we have now incorporated additional citations.
> > >
> > > > T2: Line 89: "Surface-form-consistent transformations" is an uncommon term in the literature, so it would be best to paraphrase this in layman's terms.
> > >
> > > Thanks for the suggestion! We updated the definition of "surface-form-consistent transformations" with clearer language examples: “… before rendering facts into documents. Specifically, people receive person names (Geoffrey Hinton → Caleb Ardent), cities receive city names (Toronto → Metrovale), and derived names maintain consistency (University of Toronto → University of Metrovale, not University of Grandvale).”
> > >
> > > > T3: Figure 5: Instead of breaking up the plots by expected random walk distance, it would be good to do a line plot.
> > >
> > > Our main purpose with Fig. 5 is to demonstrate the knowledge advantage gap at different difficulty levels and show how knowledge augmentation (i.e., with additional content) can close this gap. The figure shows that content consistently helps with navigation across all difficulty levels, as the agent better understands the entities it has navigated through. We chose to use defined difficulty buckets rather than a continuous scale measured by expected random walk distance because, while expected random walk distance serves as a useful proxy for difficulty, difficulty itself is inherently difficult to measure on a continuous scale. The discrete buckets make it easier to observe how the knowledge advantage gap changes as difficulty increases.

---

### Author Response · Authors · 2025-12-02
**All Reviewer and AC Response**

We thank all the reviewers for acknowledging the contributions of our work and providing insightful comments and suggestions! We are glad to see that the reviewers found:

1. **The problem formulation and framework are novel and well-motivated** (raJh, 2dyf, HETT), with SynthWorlds as "the right next step" (raJh) for synthetic evaluation, capturing real-world knowledge complexity.

2. **The framework design is sound, scalable, and controlled** (2dyf, HETT, Ra5g) with detailed automated construction, parallel task design maintaining identical reasoning structures, and controlled difficulty.

3. **The evaluation is rigorous** (Ra5g, HETT), "yielding credible, interpretable results".

4. **The presentation to be detailed and well-written** (raJh, Ra5g).


We have carefully considered, addressed, and responded to all comments and concerns provided by the reviewers (any line references are referring to the original submission). In response to reviewer concerns, we have substantially strengthened our work by:

1. **Incorporating significant additional experiments and analysis**, including an additional 40K+ evaluations across SM and RM tasks, models, and baseline conditions on 4 open-source models (gpt-oss-20b, gpt-oss-120b, kimi-k2-instruct, kimi-k2-thinking), yielding insights on model scaling and reasoning post-training impacts on SynthWorlds' performance. Our updated results reiterate/provide additional evidence…

2. **Validating performance in SynthWorld correlates with existing reasoning benchmarks**, demonstrating that our tasks capture broader reasoning capabilities while being automatic and scalable.

3. **Clarifying SynthWorld's positioning** as not only being able to generate high-quality reasoning benchmarks synthetically but, more importantly, provide complementary insights, enabling direct comparison of model behavior in novel vs. familiar environments through parallel RM/SM construction.

4. **Articulating extensibility** to domain-specific KGs, alternative domains (mathematical reasoning, code generation), mixing multiple subgraphs from different-domain KGs before document generation, and KG construction from unstructured text (e.g., Wikipedia) as important future work (Section 6).

**We believe these revisions address all major concerns while preserving the core novelty, motivation, framework, and evaluation that reviewers found valuable.**

---

### Meta-Review · Area_Chair_NpnG · 2026-01-05

**Summary:**

The authors propose SynthWorlds, a method for mirrored “real-mapped” (RM) vs “synthetic-mapped” (SM) benchmark construction to disentangle reasoning from parametric knowledge, and introduce the Knowledge Advantage (KA) gap to quantify reliance on memorized facts. Reviewers were generally positive about the motivation, clarity, and the controlled parallel-world design, and found the pipeline and evaluations credible. The main concerns were (i) an initially small set models, (ii) dependence on Wikidata/knowledge graphs and questions about robustness if the benchmark becomes widely adopted, (iii) limited validation / correctness-by-design given LM-generated text, and (iv) whether the metric is “surprising”. The rebuttal added substantial new experiments and clarifications, addressing the key generalizability concerns and improving methodological clarity, leaving mostly scope/positioning concerns as outstanding. I, therefore, believe this paper would make a solid contribution to the conference.

**Reviewer Concerns:**

Addressed by rebuttal:

Generalizability: authors added multiple open-source models, including scaling and post-training comparisons; much stronger evidence that KA persists beyond two proprietary models.
Clarity: clarified difficulty proxies (hops/motifs; expected random-walk distance), “near random,” and how error bars are computed.
RM vs SM retrieval differences: explained LM-indexed retrieval effects (HippoRAG2) and added discussion.
Failure modes: added qualitative analysis showing knowledge leakage/interference in RM and cue/retrieval bottlenecks in SM
Regenerability vs memorization: clarified stochastic pipeline and how new SM variants can be generated.

Perhaps still missing
applicability beyond Wikidata
adding more than two tasks/domains; but could be left for future work

**Reviewer Scores:**

raJh (8, confidence 5): likely unchanged at 8: most of their concerns were addressed.

2dyf (6, confidence 4): likely +1 to 7 : rebuttal added correlation discussion and clarified positioning

HETT (4, confidence 4): likely +2 to 6  main critique is directly addressed with open-model experiments.

Ra5g (6, confidence 3): likely unchanged at 6: open-model additions address their key weakness; other questions (retrieval ablations/figures) might remain open.

---

### Decision · Program_Chairs · 2026-01-26

Accept (Poster)